# Goal-Guided Efficient Exploration via Large Language Model in Reinforcement Learning

## Abstract

Real-world decision-making tasks typically occur in complex and open environments, posing significant challenges to reinforcement learning (RL) agents' exploration efficiency and long-horizon planning capabilities. A promising approach is LLM-enhanced RL, which leverages the rich prior knowledge and strong planning capabilities of LLMs to guide RL agents in efficient exploration. However, existing methods mostly rely on frequent and costly LLM invocations and suffer from limited performance due to the semantic mismatch. In this paper, we introduce a Structured Goal-guided Reinforcement Learning (SGRL) method that integrates a structured goal planner and a goal-conditioned action pruner to guide RL agents toward efficient exploration. Specifically, the structured goal planner utilizes LLMs to generate a reusable, structured function for goal generation, in which goals are prioritized. Furthermore, by utilizing LLMs to determine goals' priority weights, it dynamically generates forward-looking goals to guide the agent's policy toward more promising decision-making trajectories. The goal-conditioned action pruner employs an action masking mechanism that filters out actions misaligned with the current goal, thereby constraining the RL agent to select goal-consistent policies. We evaluate the proposed method on Crafter and Craftax-Classic, and experimental results demonstrate that SGRL achieves superior performance compared to existing state-of-the-art methods.

## 1 Introduction

Reinforcement learning (RL) has achieved impressive success in addressing challenging decision-making tasks across a wide range of domains, such as Atari games (Hessel et al., 2018; Vinyals et al., 2019), robotics (Brunke et al., 2022; Haarnoja et al., 2024), and natural language processing (Padmakumar & Mooney, 2021; Ouyang et al., 2022). However, these remarkable successes have mostly occurred in environments characterized by closed, predefined tasks with clear goals and immediate feedback, which fail to capture the complexity and dynamics of the real world (Team et al., 2021; Cai et al., 2023). In recent years, increasing attention has been devoted to decision-making in open-world environments such as Minecraft (Fan et al., 2022; Lin et al., 2022) and Crafter (Hafner, 2022; Moon et al., 2023), which pose significant challenges in generalization, deep exploration, long-term decision-making, and reasoning. Consequently, solving decision-making problems in open-world environments is widely recognized as a pressing and significant challenge.

Recently, LLM-enhanced RL has been regarded as a promising direction for addressing decision-making challenges in open-world environments (Liu et al., 2023; Zhou et al., 2024; He et al., 2024; Schoepp et al., 2025) due to the remarkable capabilities of LLMs in various decision-making and reasoning tasks. A variety of methods, such as using LLMs as decision-makers (Shinn et al., 2023; Carta et al., 2023; Gaven et al., 2024) or as skill planners (Ichter et al., 2022; Zhang et al., 2023; Lin et al., 2023; Yang et al., 2025), have emerged and significantly improved sample efficiency and generalization, thereby enhancing performance. However, these methods still struggle to fully unlock and leverage the planning capabilities of LLMs, as they either force the models to perform fine-grained planning (an area where they are not particularly adept) or fail to effectively coordinate the relationship between the LLM and RL agent components. In order to harness the capabilities of LLMs, recent studies have explored a more direct and effective approach that uses an LLM to generate high-level goals for guiding exploration in the RL agent (Du et al., 2023; Zheng et al., 2024; Shukla et al., 2024; Zhang & Lu, 2024). To mention a few, ELLM (Du et al., 2023) leverages

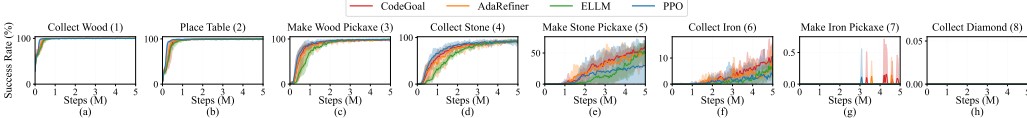

Figure 1: Success rate curves for 8 main achievements on Craftax-Classic, ordered from left to right by achievement depth from 1 to 8. A complete and more intuitive version is shown in Figure 10 of the Appendix C.

a pre-trained LLM to generate potential exploration goals, guiding the RL agent to explore towards potentially useful targets. AdaRefiner (Zhang & Lu, 2024) iteratively refines task understanding through feedback from the RL agent, thereby improving the quality of LLM-generated goals and fostering more effective collaboration between LLM and RL agent. However, due to the reliance on frequent and intensive LLM invocations, these methods suffer from low practical utility and poor computational efficiency.

Inspired by Ma et al. (2024); Xie et al. (2024), in which LLMs generate reward-shaping code to provide immediate feedback to RL agents and achieve strong performance on several benchmarks, we hypothesize that structured, goal-specifying code can also serve as a stable and executable interface between LLMs and RL agents, thereby enabling effective long-horizon exploration. Thus, we conducted preliminary experiments based on the simple idea of leveraging LLM-generated code to guide the exploration of RL agents, which we term CodeGoal. Figure 1 presents the success rate of CodeGoal in comparison with PPO, as well as ELLM (Du et al., 2023)[1] and AdaRefiner (Zhang & Lu, 2024)[2] on the Crafter-Classic benchmark. From Figure 1, we can observe that compared with ELLM (Du et al., 2023), AdaRefiner (Zhang & Lu, 2024), and PPO, CodeGoal achieves competitive success rates when unlocking key achievements, demonstrating that using code-generated goals to guide RL agents is a viable approach.

It is worth noting that the experimental results in Figure 1 also reveal two key limitations: (1) the agent fails to quickly unlock deeper achievements such as *Collect Diamond*; (2) integrating textual goals into the RL agent's decision-making process yields limited immediate benefits, evidenced by the apparent ineffectiveness of goal guidance within the first 2M steps. This motivates us to propose a novel LLM-based goal-conditioned RL approach, which first leverages LLMs to generate a parameterized, well-structured, and reusable goal-generation function, and then utilizes the LLMs to optimize both the parameters of this function and the goal-conditioned policy constraints, thereby encouraging RL agents to explore effectively in open-world environments.

The main contributions are summarized as follows:

- We propose Structured Goal-guided Reinforcement Learning (SGRL), an LLM-enhanced RL method that constructs a structured goal-generation function and dynamically adjusts both goal priority weights and goal-conditioned action constraints, thereby significantly improving the RL agent's exploration efficiency and overall performance, while maintaining low LLM invocation frequency and minimal input token consumption per call.

- We develop a structured goal planner that leverages the LLM to construct a reusable, structured goal-generating function that selects forward-looking goals and dynamically adjusts their priority weights during training. Furthermore, a goal-conditioned action pruner is designed to filter out actions misaligned with the goal, thereby constraining agents to select goal-consistent policies.

- Extensive experimental results in the challenging open-world environments Crafter and Craftax-Classic demonstrate that SGRL consistently outperforms or matches existing LLM-enhanced RL methods across multiple metrics, including success rate, total score, cumulative reward, and achievement depth.

---

[1] using DeepSeek-V3 model as goal generator, queried every 200 steps under training time and cost constraints.

[2] using DeepSeek-V3 model as both adapter and decision LLM, queried every 200 steps under training time and cost constraints.

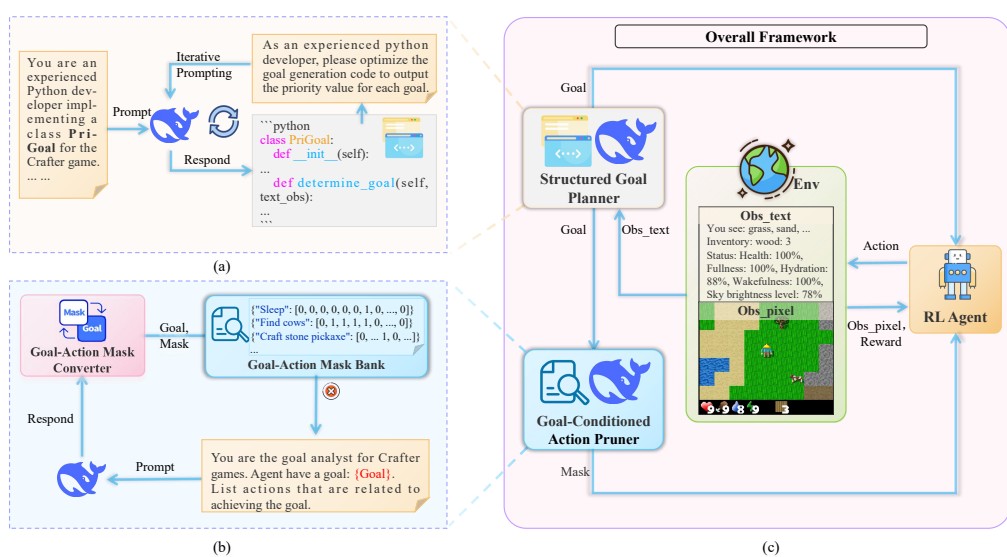

Figure 2: (a) Structured Goal Planner: generates goals with priority weights based on environment states and distilled task knowledge; (b) Goal-Conditioned Action Pruner: filters invalid or irrelevant actions based on current goals; (c) The overall framework of SGRL.

## 2 PRELIMINARY

### 2.1 GOAL-CONDITIONED REINFORCEMENT LEARNING

A goal-conditioned reinforcement learning (GCRL) (Liu et al., 2022) problem can be formulated as a goal-augmented MDP, denoted as a tuple $< \mathcal{S}, \mathcal{A}, \mathcal{G}, p_g, \phi, \mathcal{P}, r, \rho_0, \gamma >$, where $\mathcal{S}$, $\mathcal{A}$ and $\mathcal{G}$ denote the state space, action space and goal space, respectively; $p_g$ and $\rho_0$ denote the desired goal distribution and initial state distribution, respectively; $\phi : \mathcal{S} \to \mathcal{G}$ is a mapping function that maps the state to a specific goal; $\mathcal{P} : \mathcal{S} \times \mathcal{A} \to \mathcal{S}$ represents the state transition function; $r : \mathcal{S} \times \mathcal{A} \times \mathcal{G} \to \mathbb{R}$ is the reward function; and $\gamma$ is the discount factor.

At each timestep $t$, given the state $s_t$ and the desired goal $g$, the agent takes an action $a_t$ according to its policy $\pi(a_t \mid s_t, g)$ to interact with the environment and then receives the next state $s_{t+1}$ and reward $r_{t+1}$. The agent's goal is to maximize the expected sum of discounted rewards over the state-goal distribution:

$$\mathcal{J}(\pi) = \mathbb{E}_{\substack{g \sim p_g, s_0 \sim \rho_0, \\ a_t \sim \pi(\cdot|s_t, g)}} \left[ \sum_t \gamma^t r\left(s_t, g, a_t\right) \right]. \tag{1}$$

### 2.2 LLM AS GOAL PLANNER

Formally, given a natural language task description $\omega \in \Omega$, a sequence of goals is generated by an LLM-based goal planner. At each planning step $h$, the LLM takes the state-goal history $\tau_h = \{s_1, g_1, \ldots, s_{h-1}, g_{h-1}, s_h\}$ as input and generates the next goal:

$$g_h \sim \pi_{\text{LLM}}(\cdot \mid \tau_h, \omega),$$

where the goal $g_h$ is expressed in natural language and belongs to the language space $\mathfrak{L}$, which contains possible linguistic expressions such as sentences or phrases describing actionable intentions. For practical purposes, a task-specific subset $\mathcal{G} \subseteq \mathfrak{L}$ is often considered to restrict the goal space to valid and executable instructions. Subsequently, the agent choose a policy $\pi$ conditioned on both the current environment state $s_h \in \mathcal{S}$ and the goal $g_h$:

$$a_h \sim \pi(\cdot \mid s_h, g_h),$$

where $a_h \in \mathcal{A}$ denotes the agent's action at time step $h$.

## 3 METHOD

This section develops a novel LLM-enhanced RL method called Structured Goal-guided Reinforcement Learning (SGRL). Figure 2 (c) shows an overview of the SGRL framework, which consists of three key components: (1) a structured goal planner, which receives textual observations from the environment and provides forward-looking goals; (2) a goal-conditioned action pruner, which takes the goal from the structured goal planner as input and generates an action mask; and (3) an RL agent, which executes actions according to a policy conditioned on the current environmental state, goal and most recent reward. The details are introduced in the next subsections.

### 3.1 STRUCTURED GOAL PLANNER

The core task of the structured goal planner is to generate the goal function and optimize the goal priority weights by LLMs through task-specific prompting, as illustrated in Figure 2 (a). Specifically, given a basic task introduction, its rules, and a few code examples, the LLM generates an executable, structured function for goal generation according to task-specific and environmental state-related prompting. Then, in just a few iterations and optimizations, we can obtain a reusable, structured goal generation function with priority weights, which can provide forward-looking goals to guide RL agent exploration efficiency. Furthermore, in actual execution, the structured goal planner adjusts the weights of goals based on the agent's unlocked achievements at different training stages.

Formally, at each timestep $t$, the structured goal planner constructs a function $\{(g_t^i, w_t^i)\}_{i=1}^k \sim \phi(s_t)$, which maps the current state $s_t \in \mathcal{S}$ to a set of $k$ candidate goals $g_t$ with priority weights $w_t$. Here, $g_t^i \in \mathcal{G}$ denotes the $i$-th goal from the goal space $\mathcal{G}$, and $w_t^i \in [0, 1]$ is the normalized priority value for that goal, satisfying $\sum_{i=1}^k w_i = 1$. These priority values serve to rank candidate goals to guide the agent's exploration toward the most relevant and achievable objectives. The goal planning function $\phi(s)$ is constructed directly by LLM with task-specific and current state-related prompts, unlike the method in which the LLM as a goal planner that generates goals $g_t \sim \pi_{\text{LLM}}(\cdot \mid \tau_t, \omega)$ without explicitly defining a reusable goal-generating function. It is worth noting that the structured goal generation method can provide semantically forward-looking goals to improve the RL agent's exploration efficiency, while maintaining a low LLM invocation frequency and minimal input token consumption per call.

### 3.2 GOAL-CONDITIONED ACTION PRUNER

The core task of the goal-conditioned action pruner is to constrain the agent to select goal-consistent policies, as shown in Figure 2 (b). Specifically, we first leverage the LLM with rich prior knowledge to filter from the candidate action set those actions aligned with the current goal. This filtering is implemented via a goal-action masking mechanism. In practice, we construct a goal-action mask bank to store goal-mask pairs, eliminating redundant LLM queries when the same goal reappears.

Formally, at each timestep $t$, the action pruner produces a binary mask $m(g_t^i) \in {0, 1}^{|\mathcal{A}|}$ base on the candidate goals $\{g_t^1, \ldots, g_t^k\}$, which are obtained from the LLM or the goal-action mask bank. Each element of $m(g_t^i)$ indicates whether the action is relevant to the goal $g_t^i$. Then, the action pruner extracts actions relevant to any given goal $g_t^i$ in an element-wise fashion as follows:

$$M = \max_{i=1,\ldots,k} m(g_t^i). \tag{2}$$

However, it is worth noting that strictly adhering to goal guidance at all times is not always optimal, especially when using a general-purpose LLM without fine-tuning or domain-specific expertise. Therefore, we introduce a masking coefficient that grants the RL agent some autonomy in decision-making. Specifically, we design a three-phase cosine annealing schedule for the exploration coefficient $\xi \in [0, 1]$, which gradually adjusts the strength of action masking throughout training. Specifically:

$$\xi(t) = \begin{cases} \frac{1}{2}\left(1 + \cos\left(\frac{t}{0.4T} \cdot \pi\right)\right), & 0 \le t < 0.4T \\ \frac{1}{2}\left(1 - \cos\left(\frac{t-0.4T}{0.4T} \cdot \pi\right)\right), & 0.4T \le t < 0.8T \\ 1.0, & t \ge 0.8T \end{cases} \tag{3}$$

Then, the coefficient $\xi$ is used to stochastically relax the mask via a Bernoulli sampling mechanism:

$$\tilde{M}^j = \max\left(M^j, \text{Bernoulli}(\xi \cdot (1 - M^j))\right), \tag{4}$$

where $M^j$ is the the $j-th$ element of $M$, $j = 1, \ldots, |\mathcal{A}|$. Equation (4) allows masked-out actions to be sampled with a small probability, thereby mitigating policy rigidity caused by inaccurate masks or shifts in environmental dynamics. Notably, we provide the performance comparison of algorithms with different masking strategies in Appendix F.

## 3.3 RL Agent

In this subsection, we introduce how the RL agent makes decisions based on its current state, as well as goals and action mask constraints.

First, the goal set with priority weights $\{(g_t^i, w_t^i)\}_{i=1}^k$ is encoded into a goal embedding vector $g_t^{\text{emb}}$ via the encoder network. The embedding vector $g_t^{\text{emb}}$ is concatenated with the state $s_t$ to form an augmented state $[s_t; g_t^{\text{emb}}]$, which serves as the input to the policy network $\pi_\theta(\cdot \mid s_t, g_t^{\text{emb}})$. Then, based on the action mask $\tilde{M}$, the raw logits output can be obtained to enforce action feasibility:

$$\texttt{logits} = \texttt{actor\_logits} \odot \tilde{M} + (1 - \tilde{M}) \cdot (-C), \tag{5}$$

where $\odot$ denotes element-wise multiplication, and $C \gg 0$ is a sufficiently large constant (e.g., $10^6$) that suppresses the logits of invalid actions to near negative infinity.

In practice, the policy $\pi_\theta(a_t \mid s_t, g_t^{\text{emb}})$ of the RL agent is updated using the PPO algorithm (Schulman et al., 2017) with state augmentation by optimizing the following objective:

$$\mathcal{L}_\pi = \mathbb{E}_{\substack{s_t, g_t^{\text{emb}} \sim \mathcal{D}, \\ a_t \sim \pi_{\text{old}}(\cdot \mid s_t, g_t^{\text{emb}}), \\ s_{t+1} \sim \mathcal{P}(\cdot \mid s_t, a_t)}} \left[ \min\left( \frac{\pi_\theta(a_t \mid s_t, g_t^{\text{emb}})}{\pi_{\text{old}}(a_t \mid s_t, g_t^{\text{emb}})} \hat{A}_t, \text{clip}\left( \frac{\pi_\theta(a_t \mid s_t, g_t^{\text{emb}})}{\pi_{\text{old}}(a_t \mid s_t, g_t^{\text{emb}})}, 1 - \epsilon, 1 + \epsilon \right) \hat{A}_t \right) \right], \tag{6}$$

where $\hat{A}_t$ is the estimated advantage function, $\epsilon$ is the clipping parameter, and $\mathcal{D}$ denotes the replay buffer or on-policy rollout distribution. The mask $\tilde{M}$ affects both the behavior policy $\pi_{\text{old}}$ and the updated policy $\pi_\theta$, ensuring consistency between sampling and optimization.

## 4 Related Works

### 4.1 Open-World Environments

Open-world environments (Team et al., 2021; Cai et al., 2023) are inherently challenging due to requirements for generalization, exploration, multi-objective optimization, and long-horizon planning and reasoning (Hafner, 2022; Wang et al., 2023). There are three main approaches for applying RL in open-world environments in the existing literature. One approach is hierarchical reinforcement learning (Hutsebaut-Buysse et al., 2022), which simplifies complex decision-making processes by constructing a multi-level subtask structure. However, due to the inherent limitations of reinforcement learning algorithms in planning and reasoning, the generalization and long-term decision-making capabilities of these methods are still constrained. Another approach is model-based RL (Moerland et al., 2023; Walker et al., 2023), which learns an explicit environment dynamics model to enable more sample efficient through simulated rollouts. However, these methods require learning an accurate world model, which results in significantly higher computational overhead, particularly in open-world environments with high-dimensional observations. With the rapid development of LLMs, recent studies have explored integrating LLMs into RL pipelines (Zhou et al., 2024; Schoepp et al., 2025). Leveraging their extensive prior knowledge, reasoning capabilities, and strong generalization, LLMs have been employed to provide high-level planning for RL agents (Du et al., 2023; Zhang & Lu, 2024; Prakash et al., 2023; Yan et al., 2025), which are discussed in detail in the following subsection.

### 4.2 LLM-Enhanced RL

LLM-enhanced RL (Zhou et al., 2024; Schoepp et al., 2025), in which LLMs are employed as goal generators or policy selectors, with the core idea being to exploit their extensive prior knowledge for

more effective task decomposition and decision-making. To mention a few, SayCan (Ichter et al., 2022), BOSS (Zhang et al., 2023) and When2Ask (Hu et al., 2024) utilize LLMs as skill planner to construct high-level plans or feasible skill sequences base on natural language instructions or task descriptions. However, these methods rely on assumed access to pretrained skills, and the generated plans lack structured representations, limiting their scalability and adaptability. Furthermore, some works (Hu & Sadigh, 2023; Prakash et al., 2023; Yan et al., 2025) take a more direct straightforward approach, utilizing LLMs as policy teacher. Due to the fact that these methods either rely on a library of pretrained skills for high-level decision-making or depend on the LLMs' reasoning and language-to-action capabilities for low-level guidance, they typically require significant computational resources or extensive pretraining infrastructure. In addition, ELLM (Du et al., 2023), AdaRefiner (Zhang & Lu, 2024), LLMV-AgE (Chi et al., 2025) and Ruiz-Gonzalez et al. (2024) employ LLMs as goal generators to produce semantic subgoals that guide exploration. Unfortunately, due to their reliance on frequent and intensive LLM invocations, these methods suffer from low practical utility and poor computational efficiency.

## 5 EXPERIMENTS

In this section, experiments are conducted on two open-world RL benchmarks: Crafter (Hafner, 2022) and Craftax-Classic (Matthews et al., 2024). The experiments aim to answer the following questions: 1) How does the exploration efficiency of SGRL compare with existing LLM-enhanced RL methods? 2) How do goal-conditioned policy constraints contribute to the performance of SGRL?

To answer these questions, we compare SGRL against the following algorithms: ELLM (Du et al., 2023), which generates goals to guide agent exploration through online queries of an LLM; AdaRefiner (Zhang & Lu, 2024), which enhances the quality of the LLM-generated goals by refining the prompts; and PPO (Schulman et al., 2017), which is a pure RL algorithm that does not involve an LLM. Notably, for the specific hyperparameter settings of the PPO algorithm, we follow the *Stable-Baselines3* (Raffin et al., 2021)[3] implementation for Crafter, and follow the official in *Craftax* benchmark (Matthews et al., 2024)[4] implementation for Craftax-Classic. In addition, human expert performance (Hafner, 2022) is included as a reference.

### 5.1 EXPERIMENTAL SETUP AND EVALUATION METRICS

#### 5.1.1 ENVIRONMENT

Crafter is a widely used benchmark for open-world environments, evaluating agents on generalization, exploration, and long-term reasoning through 22 diverse achievements. The Craftax-Classic environment re-implements Crafter in JAX. Both benchmarks feature sparse rewards, complex goal hierarchies, and open-world exploration, making them ideal for evaluating our framework's ability to provide structured semantic guidance. Further details of the environmental setup are provided in Appendix A.1.

The evaluation metrics from Matthews et al. (2024), including success rate, score, return, and achievement depth, to comprehensively assess the performance of SGRL compared to the baseline algorithms. In addition, the training speed is reported in steps per second (SPS), presented in the tables as SPS ($\times 10^2$) for readability. Further details can be found in Appendix A.2.

#### 5.1.2 COMPUTE RESOURCES

Experiments on Crafter were conducted using a single A100 GPU with 40 GB of VRAM. Experiments on Craftax-Classic were performed on a system equipped with an NVIDIA GeForce RTX 4090 (24 GB) and an Intel(R) Core(TM) i9-14900K CPU. Results for both our algorithm and baseline methods are based on the same configurations. All reported results are averaged over five random seeds and learning curves are smoothed over time.

---

[3]available at `https://github.com/DLR-RM/stable-baselines3`
[4]available at `https://github.com/MichaelTMatthews/Craftax`

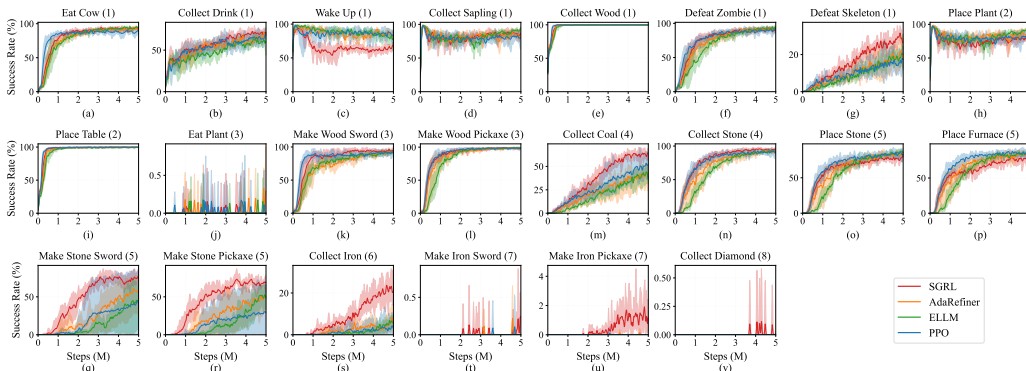

Figure 3: Success rate curves for all achievements on Craftax-Classic. Achievements are ranked based on their depth and the importance of unlocking them for subsequent tasks. Achievements ranked later have greater depth and exert a stronger influence on subsequent achievements. A more intuitive version is shown in Figure 13 in Appendix D.

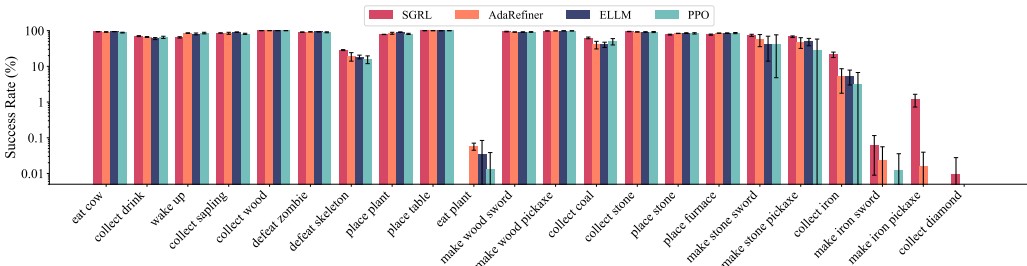

Figure 4: Success rates across all achievements on Craftax-Classic at 5M steps.

## 5.2 MAIN RESULTS

Figure 3 shows the success rate curves for all 22 achievements on Craftax-Classic. From Figure 3, we can observe that SGRL consistently achieves higher success rates than the baselines when reaching the final few achievements (see Figure 3 (q)-(v)). However, for achievements like Wake Up (see Figure 3 (c)) and Place Plant (see Figure 3 (h)), which do not facilitate later exploration, SGRL's performance plateaus once high success rates are reached. This phenomenon demonstrates that our goal-generation method produces farsighted objectives, enabling SGRL to transcend short-term rewards and maintain a stable and coherent policy in long-horizon decision-making tasks. Moreover, as shown in Figure 3 (v), SGRL successfully unlocks the *Collect Diamond* achievement shortly after 3.7M steps, whereas AdaRefiner, ELLM, and PPO fail to achieve it even by 5M steps, demonstrating SGRL's effectiveness in enhancing exploration efficiency. Figure 4 provides a more direct visualization, which clearly highlights SGRL's advantage in unlocking late-stage achievements, confirming its capability for effective long-horizon planning. More experimental results can be found in Appendix D.

| Method | Score(%) | Reward | Achievement Depth | SPS ($\times 10^2$) |
|---|---|---|---|---|
| Human | $50.5 \pm 6.8$ | $14.3 \pm 2.3$ | 8 | - |
| SGRL | $\mathbf{33.8} \pm 1.5$ | $\mathbf{13.0} \pm 0.3$ | **8** | 18.5 |
| AdaRefiner | $28.5 \pm 2.3$ | $12.3 \pm 0.9$ | 7 | 0.3 |
| ELLM | $28.4 \pm 2.5$ | $12.2 \pm 1.0$ | 6 | 0.9 |
| PPO | $24.8 \pm 5.7$ | $11.9 \pm 1.1$ | 6 | 135.3 |

Table 1: Performance of SGRL and baseline methods on Craftax-Classic at 5M steps.

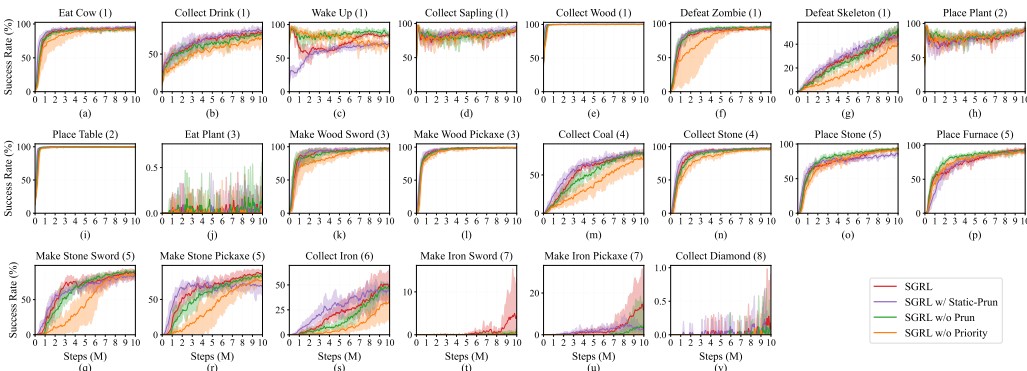

Figure 5: Ablation Studies. Success rate curves for all achievements on Craftax-Classic. Achievements are ranked based on their depth and the importance of unlocking them for subsequent tasks. Achievements ranked later have greater depth and exert a stronger influence on subsequent achievements. A more intuitive version is shown in Figure 20 in Appendix E.1.

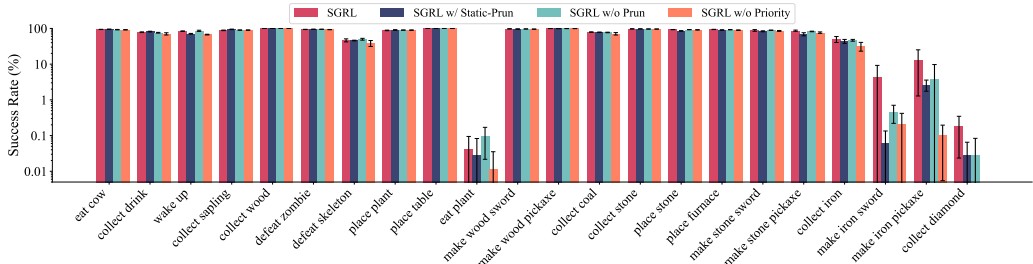

Figure 6: Ablation Studies. Success rates across all achievements on Craftax-Classic at 10M steps.

Table 1 summarizes the overall performance of SGRL and baseline methods on Craftax-Classic at 5M steps. As demonstrated in Table 1, SGRL outperforms AdaRefiner, ELLM, and PPO in terms of overall score and achievement depth on Craftax-Classic. In addition, SGRL maintains a high reward level, yet does not exhibit a significant advantage in this regard. We attribute this to the misalignment between reward magnitude and exploration depth in the environment, which forces a trade-off between completing simple, reliably rewarded achievements and pursuing more challenging but potentially high-impact ones. This is evidenced by human performance: experts achieve very high scores without a corresponding increase in total reward. Further, SGRL requires only minimal LLM invocation, resulting in faster training speed. Moreover, to better illustrate the performance of the proposed method, we provide results for various algorithms in the Crafter environment, which can be found in Appendix D.

## 5.3 ABLATION STUDY

### 5.3.1 ABLATION VARIANTS

To evaluate the contributions of each component of SGRL, we conducted ablation studies using three variants of SGRL on Craftax-Classic as follows: (1) SGRL w/ Static-Prun: This variant retains the action pruning mechanism but replaces the adaptive pruning coefficient with a static masking scheme; (2)SGRL w/o Prun: This variant removes the goal-conditioned action pruner entirely; (3)SGRL w/o Priority: This variant removes the priority assignment for goals.

### 5.3.2 ABLATION ANALYSIS

Figure 5 shows the success rate curves of SGRL and its ablation variants for all 22 achievements on Crafter. From Figure 5, we can observe that SGRL significantly outperforms all variants on most of the deeper achievements after approximately 8M steps, indicating that both components of

our algorithm contribute substantially to its performance. Moreover, it is noteworthy that SGRL w/ Static-Prun, a variant of SGRL with strict action masking, exhibits strong early performance but is ultimately outperformed by SGRL in later stages, suggesting that over-reliance on the LLM may lead the agent to converge to suboptimal policies (see Figure 5 (c), (o), (p) and (q)).

Figure 6 and Table 2 present the success rates, scores, rewards, and achievement depth across all 22 achievements in the Crafter environment at 10M steps. The results demonstrate that SGRL achieves clear advantages on long-horizon tasks such as *Make Iron Pickaxe* and *Col-*

| Methd | Score (%) | Reward | Achievement Depth |
|---|---|---|---|
| SGRL | **43.9** ± 2.6 | **14.9** ± 0.4 | 8 |
| SGRL w/ Static-Prun | 38.5 ± 1.9 | 14.2 ± 0.4 | 8 |
| SGRL w/o Prun | 40.0 ± 2.1 | 14.7 ± 0.2 | 8 |
| SGRL w/o Priority | 35.3 ± 1.6 | 13.9 ± 0.6 | 7 |

Table 2: Ablation results on Craftax-Classic at 10M steps.

*lect Diamond*, which require sequential planning and tool construction. Notably, the design of goal prioritization plays a critical role in enabling SGRL to rapidly unlock deep achievements like *Collect Diamond* by guiding the agent to focus on high-value, forward-looking goals during exploration. More experimental results are in Appendix E.1.

Furthermore, for an in-depth analysis of SGRL's superiority, Figure 7 presents the goals with priority weights on Craftax-Classic. For clarity, a higher-level goal is visualized only from the point at which it is assigned a non-zero weight. The figure shows that SGRL consistently assigns a small priority weight to more forward-looking goals compared to the ablation algorithms at earlier stage, which motivates its agent to begin exploring more challenging achievements sooner. We hypothesize that this is the primary driving force behind SGRL's rapid attainment of superior performance. For more detailed results, refer to Appendix E.2, where we present the heatmap of goals with priority weights and provide a detailed analysis.

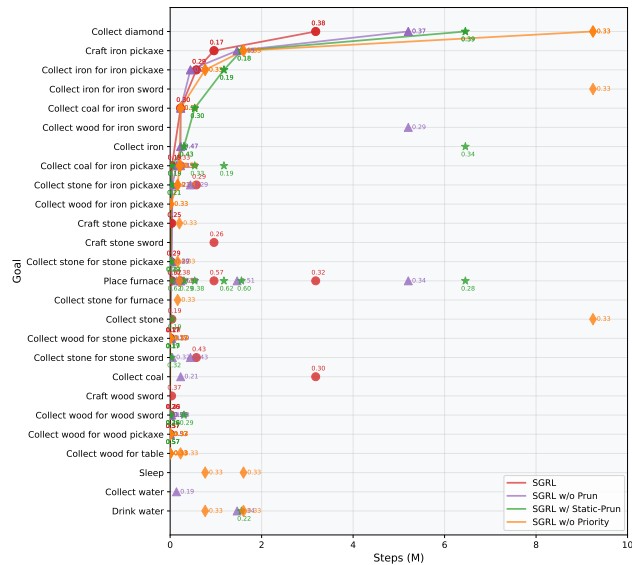

Figure 7: Ablation Studies. Goals with priority weights on Craftax-Classic, where a higher-level goal is visualized only from the point it is assigned a non-zero weight.

In summary, all ablation results collectively indicate that both the goal priority weights and action mask in our proposed SGRL play distinct roles. Specifically, assigning priorities to goals within the structured goal planner is crucial for generating reasonable and effective goals. The goal-conditioned action pruner effectively enhances the agent's exploration capability in long-horizon tasks.

## 6 CONCLUSION

This paper proposes a novel LLM-enhanced RL method called SGRL that leverages LLMs to improve RL agents' exploration efficiency and long-horizon planning capabilities in open-world environments. Specifically, we develop a structured goal planner that leverages the LLM to construct reusable goal-generating functions that select forward-looking goals and dynamically adjust their priority weights. Then, a goal-conditioned action pruner is designed to filter out actions misaligned with the goal, thereby guiding RL agents to select goal-consistent policies. Finally, extensive experimental results demonstrate that SGRL achieves superior performance compared to existing LLM-enhanced RL baselines in terms of long-horizon planning and exploration efficiency.

ETHICS STATEMENT

This work focuses on developing an LLM-enhanced RL method and does not involve human subjects, personal data, or sensitive information. The experiments are conducted on publicly available benchmark datasets and simulated environments. We believe our research raises no direct ethical concerns and may contribute positively by improving the exploration efficiency and long-horizon planning capabilities of RL methods.

REPRODUCIBILITY STATEMENT

All implementation details, including source code, hyperparameters, prompts and outputs of LLM and scripts, are provided in the appendix and supplementary material to enable full reproducibility of our results.

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

## A    ENVIRONMENTS AND EVALUATION METRICS

### A.1    ENVIRONMENTS

*Crafter (Hafner, 2022)* is a widely adopted benchmark for open-world RL, designed to assess agents' generalization, exploration, and long-term reasoning capabilities through 22 diverse achievements. These achievements are organized in a hierarchical dependency structure of up to 8 depth levels, where early-stage skills (e.g., collect wood, place table) unlock preconditions for increasingly complex tasks. The deepest and most challenging achievement (i.e., *Collect Diamond*) requires agents to master long sequences of dependencies, from crafting stone tools to mining iron and finally accessing diamonds deep underground. Crafter's procedurally generated environments further exacerbate challenges in sparse rewards, efficient exploration, and hierarchical planning, making it a strong testbed for evaluating structured goal-guided learning.

*Craftax-Classic (Matthews et al., 2024)* is a high-performance, JAX-based reimplementation of Crafter that achieves a 250x simulation speedup via vectorization and parallelization. It faithfully reproduces Crafter's core task structure, environmental dynamics, and evaluation metrics, while enabling large-scale experimentation at 1B+ environment steps within practical compute budgets.

### A.2    EVALUATION METRICS

To demonstrate the effectiveness of our algorithm, we introduce the evaluation metrics as follows:

- *Achievement Success Rate.* This metric reflects the agent's learning capability and exploration depth by measuring the probability that each predefined achievement is successfully unlocked.

- *Geometric Mean Score.* This metric reflects the balance between both easy and difficult goals. Following the official Crafter evaluation protocol (Hafner, 2022), it is defined as:

$$\text{score} = \exp\left(\frac{1}{N}\sum_{i=1}^{N}\log(1+\varrho_i)\right) - 1,$$

  where $\varrho_i \in [0, 100]$ denotes the success rate of the $i$-th achievement, and $N$ is the total number of predefined achievements.

- *Achievement Depth.* This metric measures the agent's exploration depth based on the furthest achievement it unlocks.

- *Episode Return.* This metric reports the cumulative reward received per episode.

- *Steps Per Second (SPS)*: This metric measures the number of environment steps processed per second, indicating the computational efficiency and speed of learning for each method.

## B    IMPLEMENTATION DETAILS

### B.1    LLM

We utilize `DeepSeek-R1` (Guo et al., 2025) model to generate structured goal-generating planner code. Additionally, we use `DeepSeek-V3` (Liu et al., 2024) model for selecting actions related to the defined goals. For all LLM queries, we follow the implementation of AdaRefiner (Zhang & Lu, 2024) to set the decoding parameters: a temperature of 0.5, top-p of 1.0, and a maximum token limit of 100. `DeepSeek-V3` model is employed to replicate the results of ELLM and AdaRefiner.

### B.2    TEXT EMBEDDING

For text embedding, we use `paraphrase-MiniLM-L6-v2` (Reimers & Gurevych, 2019) model as the encoder.

## B.3 PROMPT DESIGN

### B.3.1 PROMPT DESIGN FOR STRUCTURED GOAL PLANNER

In the Structured Goal Planner, the large model is employed to generate goal-generation code and to update the priorities of goals. To enable the LLM to produce high-quality code, we adopt a multi-stage prompting process:

- **Design Stage.** At this stage, the model is asked to first design the class structure and key functional modules according to the task requirements.
- **Implementation Stage.** After the design, the model is prompted to output detailed, complete Python code that follows PEP8 standards with clear logic and sufficient comments.
- **Reflection and Revision Stage.** Finally, the model is prompted to reflect on the generated code, identify potential issues, and provide corrections or optimizations.

To guide the model in updating goal priorities, the prompt additionally specifies that every 2 million steps the LLM should update the priority values of goals within the generated code. This ensures that goal selection remains dynamic and aligned with the agent's current objectives.

---

**Prompt Template for Structured Goal Planner Design**

Crafter is a 2D open-world survival game with visual input; its world is procedurally generated. Players must search for food and water, find shelter to sleep, defend against monsters, gather materials, and craft tools. Crafter's objective is to evaluate an agent's capabilities through a series of semantically meaningful achievements that can be unlocked in each playthrough—for example, discovering resources and crafting tools. Consistently unlocking all achievements requires strong generalization, deep exploration, and long-term reasoning. You are an experienced Python developer. The task is to create a key functional module of an advanced goal-generation system that can dynamically produce prioritized goals based on textual environment observations. The system must include functions for survival-need assessment, a tool-crafting tree, resource-collection configuration, and achievement tracking. You should primarily design the `OptimizedGoalGenerator` class structure and its key functional modules. The Agent will call the `determine_goal` function to obtain goals:

```python
def determine_goal(self, text_obs):
    return top_three_goal
```

Each `goal` is a dictionary of the form:

```python
{'goal':    ,
 'priority':    , }
```

The state of environmental text is represented by `text_obs`:
**Example 1:**
You see: plant, zombie, tree, grass, sand, path, stone
Inventory: wood: 1
Status: health: 11%, Fullness: 0%, Hydration: 0%, Wakefulness: 88%
Sky brightness level: 68% **Example 2:**
You see: plant, tree, grass, path, stone
Inventory:
Status: health: 99%, Fullness: 77%, Hydration: 66%, Wakefulness: 77%
Sky brightness level: 99%

Now please provide the `OptimizedGoalGenerator` class structure and its key functional modules.

```python
class OptimizedGoalGenerator:
    def __init__(self):
        ...
    def determine_goal(self, text_obs):
        ...
```

---

**Prompt Template for Structured Goal Planner Implementation**

Crafter is a 2D open-world survival game with visual input, where the world is procedurally generated. Players need to find food and water, locate a place to sleep, defend against monsters, gather materials, and craft tools. The objective of Crafter is to evaluate an agent's capabilities through a series of semantically meaningful achievements, which can be unlocked in each game session, such as discovering resources and crafting tools. Continuously unlocking all achievements requires strong generalization, deep exploration, and long-term reasoning.

As a Python expert, your task is to create an advanced goal generation system that dynamically generates prioritized goals based on environmental observation text. The system should include survival needs assessment, a crafting dependency tree, resource collection configuration, and achievement tracking.

Environment Details:

**Items:** sapling, wood, stone, coal, iron, diamond, wood_pickaxe, stone_pickaxe, iron_pickaxe, wood_sword, stone_sword, iron_sword (all with max: 9, initial: 0)

**Collectable resources:** tree, stone, coal, iron, diamond, water, grass (with required tools, output, and leaves defined)

**Placable objects:** stone, table, furnace, plant (with usage, location, and type defined)

**Craftable tools:** wood_pickaxe, stone_pickaxe, iron_pickaxe, wood_sword, stone_sword, iron_sword (with required materials, nearby crafting stations, and output quantity)

**Achievements:** collect_coal, collect_diamond, collect_drink, collect_iron, collect_sapling, collect_stone, collect_wood, defeat_skeleton, defeat_zombie, eat_cow, eat_plant, make_iron_pickaxe, make_iron_sword, make_stone_pickaxe, make_stone_sword, make_wood_pickaxe, make_wood_sword, place_furnace, place_plant, place_stone, place_table, wake_up

Environment Text Rendering Function:

```python
def render_craftax_text_describ_2(self, view_arr, index):
    (map_view, mob_map, inventory_values, status_values) = view_arr

    mob_id_to_name = ["zombie", "cows", "skeletons", "arrows"]
    block_id_to_name = ["invalid", "out of bounds", "grass",
        "water", "stone", "tree", "wood", "path",
                        "coal", "iron", "diamond", "crafting
                            table", "furnace", "sand", "lava",
                        "plant", "ripe plant"]

    text_view_values = set()

    block_names = np.vectorize(lambda x:
        block_id_to_name[x])(map_view[index])
    text_view_values.update(block_names.flatten())

    mob_ids = np.argmax(mob_map[index], axis=-1)
    mob_names = np.vectorize(lambda x: mob_id_to_name[x])(mob_ids)
    mob_mask = mob_map[index].max(axis=-1) > 0.5
    text_view_values.update(mob_names[mob_mask].flatten())
    text_view = ", ".join(text_view_values)

    inv_names = ["wood", "stone", "coal", "iron", "diamond",
        "sapling",
                    "wood pickaxe", "stone pickaxe", "iron pickaxe",
                    "wood sword", "stone sword", "iron sword"]
    text_obs_inv = ", ".join([f"{name}: {value}"
                                for name, value in zip(inv_names,
                                    inventory_values[index])
                                if value > 0])

    status_names = ["Health", "Fullness", "Hydration",
        "Wakefulness", "Sky brightness level"]
```

**Prompt Template for Structured Goal Planner Implementation (continued)**

```python
    status = ", ".join([f"{name}: {int(value / 0.09)}%"
                        for name, value in zip(status_names,
                            status_values[index])])

    text_obs = "You see: " + text_view + "\nInventory: " +
        text_obs_inv + "\nStatus: " + status
    return text_obs
```

Example Environmental Text Observations:
Example 1:
You see: plant, zombie, tree, grass, sand, path, stone
Inventory: wood: 1
Status: health: 11%, Fullness: 0%, Hydration: 0%, Wakefulness: 88%, Sky brightness level: 68%
Example 2:
You see: plant, tree, grass, path, stone
Inventory:
Status: health: 99%, Fullness: 77%, Hydration: 66%, Wakefulness: 77%, Sky brightness level: 99%
You have already designed the code architecture. Your goal is to complete this code and create an advanced goal generation system capable of dynamically generating prioritized goals based on environmental observation text.
The code architecture:
{last_llm_response}

**Prompt Template for Structured Goal Planner Reflection and Revision**

You are an expert Python developer and code reviewer for Crafter's goal generation system. Your task is to critically analyze the previously designed goal planner and provide an evaluation. For each submitted code version:

- If the code is complete, correct, and efficiently implements all required functionalities (survival needs assessment, resource collection, crafting, achievement tracking, threat handling, and goal prioritization), label it as **good**.

- If there are issues, missing features, or opportunities for optimization, label it as **bad**, and provide a clear explanation of the problems.

After evaluation, if the code is labeled **bad**, generate a fully optimized and corrected version of the code that addresses all identified issues. The optimized code should:

- Correctly handle all environmental observations and edge cases.

- Properly assess and prioritize survival needs.

- Integrate resource collection, crafting, and achievement goals with correct dependencies.

- Handle threats and defensive behaviors appropriately.

- Be modular, readable, and maintainable, following Python best practices.

Please review the following Structured Goal Planner code. Evaluate its quality: provide the label **good** if it is fully correct and functional, or **bad** if improvements are needed. For **bad** code, explain the deficiencies clearly and provide a complete, optimized version of the code that fixes all issues while preserving the intended functionality.
The goal-generated code: {last_llm_response}

**Prompt Template for Updating Goal Priority Values**

You are the Goal Priority Analyst for Crafter games. Your task is to update the priority weights of goals in the goal generation code, based on the current state and action trajectory of the intelligent agent. You act as a specialized assistant whose only responsibility is to adjust priority values to improve goal selection; do not change any code logic or structure.

**Inputs provided to you:**

- **Goal generation code**: A Python code module that defines goals, their attributes, and initial priority weights. ({goal_code.py})

- **Agent state and trajectory**: A structured representation of the current state of the intelligent agent, including completed goals, actions taken, and environment status. ({agent_state.json})

**Your instructions:**

- Read the goal generation code and the agent state/trajectory.

- Update the numeric **priority weights** in the code to reflect the current importance of each goal.

- Do **not** change any function definitions, logic, or class structures. Only modify numeric values associated with goal priorities.

- Ensure the updated code is fully executable and maintains its original structure.

- Keep all interfaces unchanged so that the planner can call the updated code directly.

**Example Workflow:**

1. Load the goal generation code and parse the goals.

2. Analyze the agent's current state and past actions.

3. Determine new priority weights for each goal.

4. Replace only the priority numbers in the code with the updated values.

5. Output the complete updated Python code.

**Output Format:**

- Return the entire Python code as a single code block.

- Ensure all class and function definitions remain intact.

- Only the numeric priority values are changed.

### B.3.2 PROMPT DESIGN FOR GOAL-CONDITIONED ACTION PRUNER

In the Goal-Conditioned Action Pruner, the large model selects actions that are directly relevant to a given goal through the use of prompts, thereby enabling goal-driven behavior. To guide the model in generating goal-consistent actions, the prompt explicitly defines the meaning of each action in the action space and provides examples illustrating which actions should be chosen for specific goals. In this way, the model can effectively filter actions that align with the goal, ensuring that the agent maintains coherent and goal-directed behavior during execution.

**Prompt Template for Goal-Conditioned Action Pruner Implementation**

You are the goal analyst for Crafter games, and a goal planner provides goal guidance for game characters. The agent needs to perform one or more steps to achieve this goal, and you help the agent choose the appropriate actions to accomplish it.

**Tips:**

- The goal is something that the intelligent agent is currently capable of executing under certain conditions.

- The intelligent agent may need to move to a certain location to trigger the execution condition.

---

**Prompt Template for Goal-Conditioned Action Pruner Implementation (continued)**

- The action space consists of the following 17 actions:
    1. noop # do nothing
    2. move_left
    3. move_right
    4. move_up
    5. move_down
    6. do
    7. sleep
    8. place_stone
    9. place_table
    10. place_furnace
    11. place_plant
    12. make_wood_pickaxe
    13. make_stone_pickaxe
    14. make_iron_pickaxe
    15. make_wood_sword
    16. make_stone_sword
    17. make_iron_sword
- Among them, the action `noop` means do nothing.
- The action `do` means it can complete the following: eat plant, attack zombie, attack skeleton, attack cow, chop tree for wood, mine stone, mine coal, mine iron, mine diamond, drink water, chop grass for sapling.

**Examples:**

- Goal: {Mine Iron}
  Related actions: {move_left, move_right, move_up, move_down, do}

- Goal: {make stone pickaxe}
  Related actions: {move_left, move_right, move_up, move_down, make_stone_pickaxe}

- Goal: {sleep}
  Related actions: {sleep}

- Goal: {attack cow}
  Related actions: {move_left, move_right, move_up, move_down, do}

For each given goal, generate a set of feasible actions from the action space. Include any movements or execution actions that can reasonably help achieve the goal. Focus on feasibility rather than strict optimality.
The current goal is: goal.
Please select actions that are relevant to the goal.

---

### B.4 PPO ALGORITHM

For the specific hyperparameter settings of the PPO algorithm, we follow the *Stable-Baselines3* (Raffin et al., 2021)[5] implementation for Crafter, and follow the official in *Craftax* benchmark (Matthews et al., 2024)[6] implementation for Craftax-Classic. Since SGRL, ELLM, and AdaRefiner are all implemented based on PPO, we use the same core PPO hyperparameters, as shown in Table 3.

---

[5]Available at `https://github.com/DLR-RM/stable-baselines3`
[6]Available at `https://github.com/MichaelTMatthews/Craftax`

| Parameter | Value (Crafter) | Value (Craftax-Classic) |
|---|---|---|
| Training Steps | {1M, 5M} | {5M, 10M} |
| Learning Rate | 7e-4 | 7e-4 |
| Optimizer | Adam | AdamW |
| Batch Size | 128 | 512 |
| Number of Envs | 1 | 256 |
| Update Epochs | 16 | 4 |
| Clip Ratio | 0.2 | 0.2 |
| Discount Factor $\gamma$ | 0.97 | 0.97 |
| Entropy Coefficient | 0.01 | 0.01 |
| Value Function Coefficient | 0.5 | 0.5 |
| Activation Function | ReLU | ReLU |

Table 3: Hyperparameters for PPO

## C  ADDITIONAL PRELIMINARY RESULTS

Figures 8-10 present more detailed results of the preliminary study experiments on Craftax-Classic.

## D  ADDITIONAL MAIN RESULTS

Table 4 presents performance of SGRL and baseline methods on Crafter at 5M steps. Figures 11-17 present more detailed results of the main experiments on Crafter and Craftax-Classic. It is worth noting that:

- In the experiments on Craftax-Classic, ELLM and AdaRefiner require frequent online calls to the LLM (`DeepSeek-V3`) during training, incurring substantial computational costs and training time. Therefore, we only reproduce the results within 5M steps.
- Since the Crafter environment does not adopt the JAX framework and runs extremely slowly, we report the original results of ELLM and AdaRefiner from their papers in Table 4, rather than reproducing their experiments.

**Note:** Since ELLM and AdaRefiner require frequent online calls to the LLM (`DeepSeek-V3`) during training, they incur substantial computational costs and training time. Therefore, we only reproduce the results within 5M steps.

## E  ADDITIONAL ABLATION EXPERIMENTS

### E.1  PERFORMANCE OF ABLATION ALGORITHM

Table 5 and Figures 18- 20 present more detailed results of the ablation experiments on Craftax-Classic.

### E.2  HEATMAP OF GOAL WITH PRIORITY WEIGHTS

Figure 21 show the heatmap of the goals with priority weights generated by the structured goal planner on Craftax-Classic within 10M steps. The vertical axis on the left shows goals ranked from low to high, while the right axis (ranging from 0 to 0.8) indicates the corresponding weights.

From Figure 21, we can observe the following key phenomena:

- SGRL w/o Priority only sets collect diamond as an exploration goal after 9M steps, which likely accounts for its significantly low exploration efficiency.
- SGRL, SGRL w/ Static Pruning, and SGRL w/o Pruning—methods that assign priority weights to goals—consistently treat long-horizon, high-impact achievements (e.g., Collect stone for stone pickaxe, Collect iron) as important objectives and assign them larger priority weights.

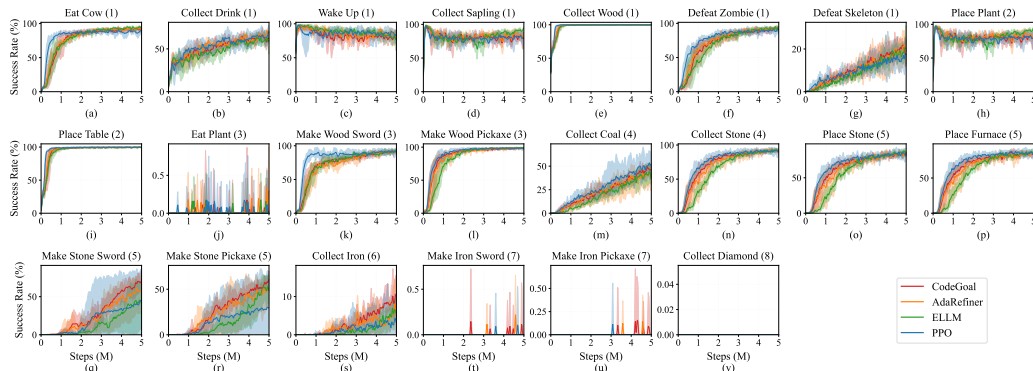

Figure 8: Preliminary Results. Success rate curves for all achievements on Craftax-Classic within 5M steps. We rank the achievements based on their depth and the importance of unlocking them for subsequent tasks. Achievements ranked later have greater depth and exert a stronger influence on subsequent achievements. A more intuitive version is shown in Figure 10.

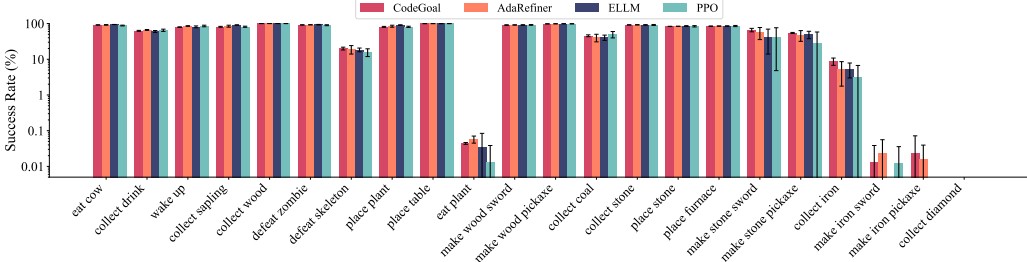

Figure 9: Preliminary Results. Success rates across all Craftax-Classic achievements at 5M steps.

- In contrast to SGRL w/ Static Pruning and SGRL w/o Pruning, SGRL consistently assigns relatively small priority weights to more forward-looking goals (e.g., Collect diamond; see the red dashed lines in the figure), which encourages the agent to begin exploring these challenging achievements earlier. We hypothesize that this adaptive prioritization strategy is the primary driver behind SGRL's superior performance.

# F ADDITIONAL PRUNER ANNEALING STRATEGIES AND IMPLEMENTATION

## F.1 PRUNER ANNEALING STRATEGIES

To dynamically adjust the agent's reliance on constraints during training, we implement several annealing strategies for $\xi$. These strategies smoothly modulate $\xi$ over training steps. These annealing schedules aim to balance the agent's adherence to the action pruner and the freedom of policy exploration.

- *Linear Annealing.* This strategy linearly anneals $\xi$ from 0 to 1, gradually reducing the influence of the pruner.

$$\xi(t) = \frac{t}{T}, \tag{7}$$

where $t$ is current training step, $T$ is total training steps.

- *Exponential Annealing.* This strategy uses exponential decay to increase $\xi$ rapidly from 0 toward 1, then asymptotically approach full freedom.

$$\xi(t) = 1 - \exp\left(-\frac{t}{\tau}\right), \tag{8}$$

where $\tau$ is the time constant of the exponential decay.

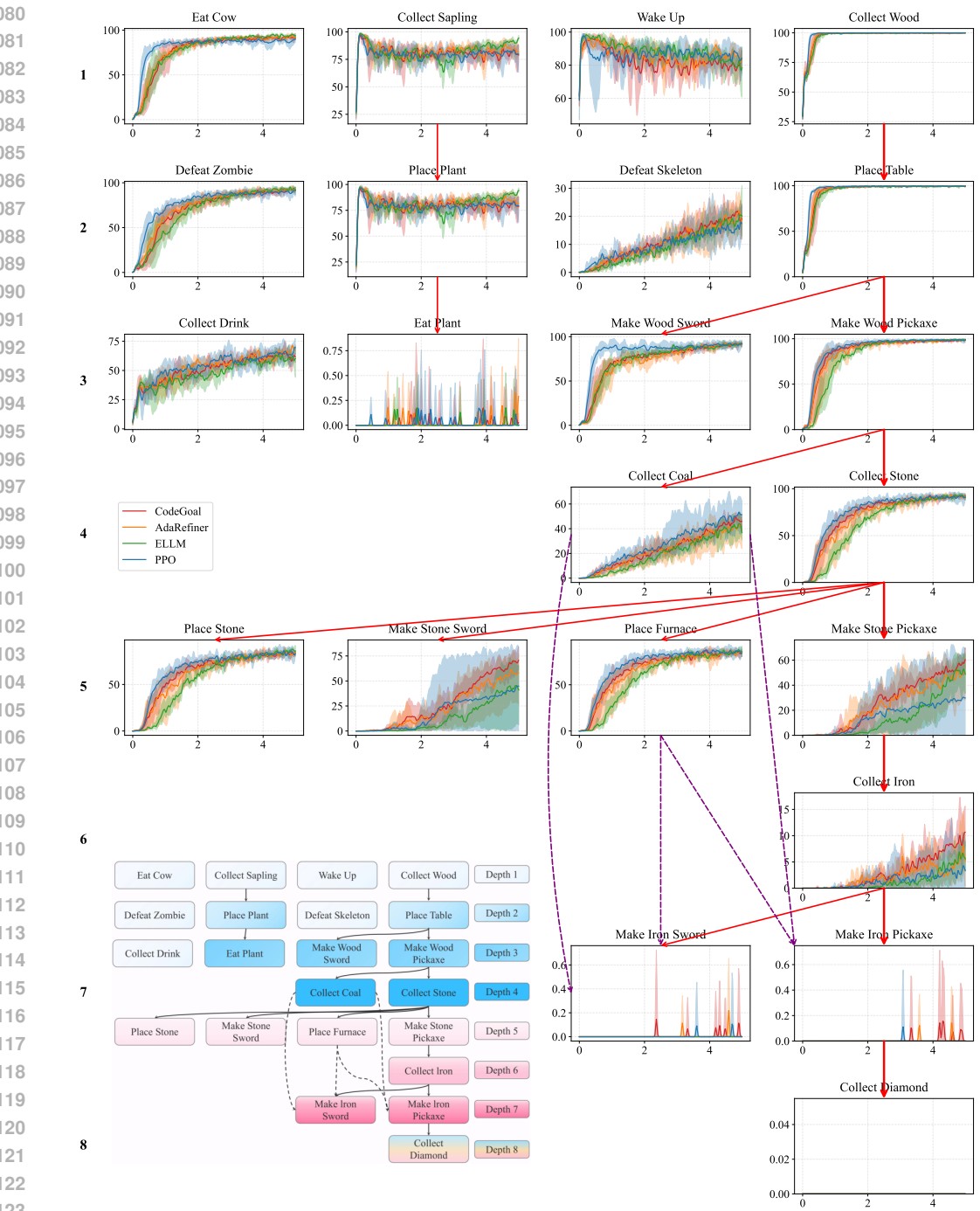

Figure 10: Preliminary Study. Success rate curves for all achievements on Craftax-Classic within 5M steps. Solid and dashed arrows indicate direct and cross-depth dependencies, respectively. The bottom-left panel visualizes the full achievement dependency graph, with achievement depth encoded by color (depth 1-8 from top to bottom).

- *Three-Stage Linear Annealing.* This strategy consists of three phases and is defined by the piecewise function:

$$\xi(t) = \begin{cases} 1 - \dfrac{t}{0.4T}, & t < 0.4T \\ \dfrac{t}{0.4T} - 1, & 0.4T \le t < 0.8T \\ 1, & t \ge 0.8T \end{cases} \cdot \tag{9}$$

- *Three-Stage Cosine Annealing.* This strategy uses a smooth cosine function for the first two stages:

$$
\xi(t) = \begin{cases}
\frac{1}{2}\left(1 + \cos\left(\frac{t}{0.4T} \cdot \pi\right)\right), & 0 \leq t < 0.4T \\
\frac{1}{2}\left(1 - \cos\left(\frac{t - 0.4T}{0.4T} \cdot \pi\right)\right), & 0.4T \leq t < 0.8T \\
1.0, & t \geq 0.8T
\end{cases} . \tag{10}
$$

In our experiments, the following naming convention is used to denote different annealing strategies applied to the SGRL: (1) *SGRL w/ Linear Ann*: SGRL equipped with linear annealing schedule (see Equation (7)); (2) *SGRL w/ Exp Ann*: SGRL with exponential annealing (see Equation (8)); (3) *SGRL w/ 3-Stage Linear*: SGRL with piecewise linear three-phase annealing (see Equation (9)); and (4) *SGRL w/ 3-Stage Cos*: SGRL with cosine-based three-phase annealing (see Equation (10)).

### F.2 EXPERIMENTAL RESULTS

Figures 22-24 present detailed results with different mask mechanism on Craftax-Classic.

As shown in Figures 22-24, the performance of SGRL with four different $\xi$ annealing strategies on Craftax-Classic is presented:

- Figure 22 displays the success rate curves across 22 achievements on Craftax-Classic, comparing four mask annealing strategies. Notably, SGRL w/ 3-Stage Cos achieves diamond collection at 3.7M steps and maintains superior performance on the most challenging achievements (Make Iron Pickaxe and Collect Diamond), as shown in Figures 22 (b)-(c). This suggests that the Three-Stage Cosine Annealing strategy enables SGRL to more effectively prioritize high-value, long-horizon objectives by adaptively balancing exploration and exploitation. In contrast, SGRL w/ Linear Ann demonstrates stronger early-stage performance, unlocking depth-7 achievements faster (see Figure 22 (a)). However, its success rate plateaus in later training phases (see Figures 22 (b)-(c)), likely due to the rigid linear decay of $\xi$, which prematurely restricts exploration and hinders adaptation to complex tasks. A success rate plot that intuitively reflects achievement depth is shown in Figure 24.
- Figure 23 presents the success rates across all 22 achievements on Craftax-Classic at different training steps. From Figure 23, we can observe that while *SGRL w/ 3-Stage Linear* and *SGRL w/ 3-Stage Cos* exhibit similar performance early on (Figure 23 (a)), the latter significantly outperforms the former in late-stage deep achievements (see Figures 23 (b)-(c)). We hypothesize that the piecewise linear transitions in 3-Stage Linear Annealing introduce abrupt changes in exploration pressure, whereas the smooth cosine modulation in 3-Stage Cosine Annealing facilitates more stable learning. Interestingly, *SGRL w/ Exp Ann* is the only variant failing to unlock Collect Diamond by 10M steps. This indicates that the rapid decay of $\xi$ in exponential annealing diminishes goal guidance too early, impairing the agent's ability to align goals with agents' actions and hindering the acquisition of complex, multi-step behaviors.

Overall, these results highlight the critical role of the annealing schedule in modulating the trade-off between goal-driven exploration and policy autonomy. The three-stage cosine strategy achieves the most effective balance, enabling sustained guidance during critical phases of skill acquisition while allowing gradual transition to policy-based control.

## G  USE OF LARGE LANGUAGE MODELS

In this work, in addition to employing Large Language Models (LLMs) as auxiliary tools for language polishing, grammar correction, and minor stylistic adjustments of the manuscript, we also utilized LLMs to generate reusable, structured goal-generation functions, adjust goal priority weights, and establish goal-conditioned policy constraints. These operations are both essential and commonplace in LLM-enhanced RL research Notably, LLMs were not involved in research ideation, experimental design, implementation, data analysis, or result interpretation. All scientific contributions and intellectual content are solely attributable to the authors.

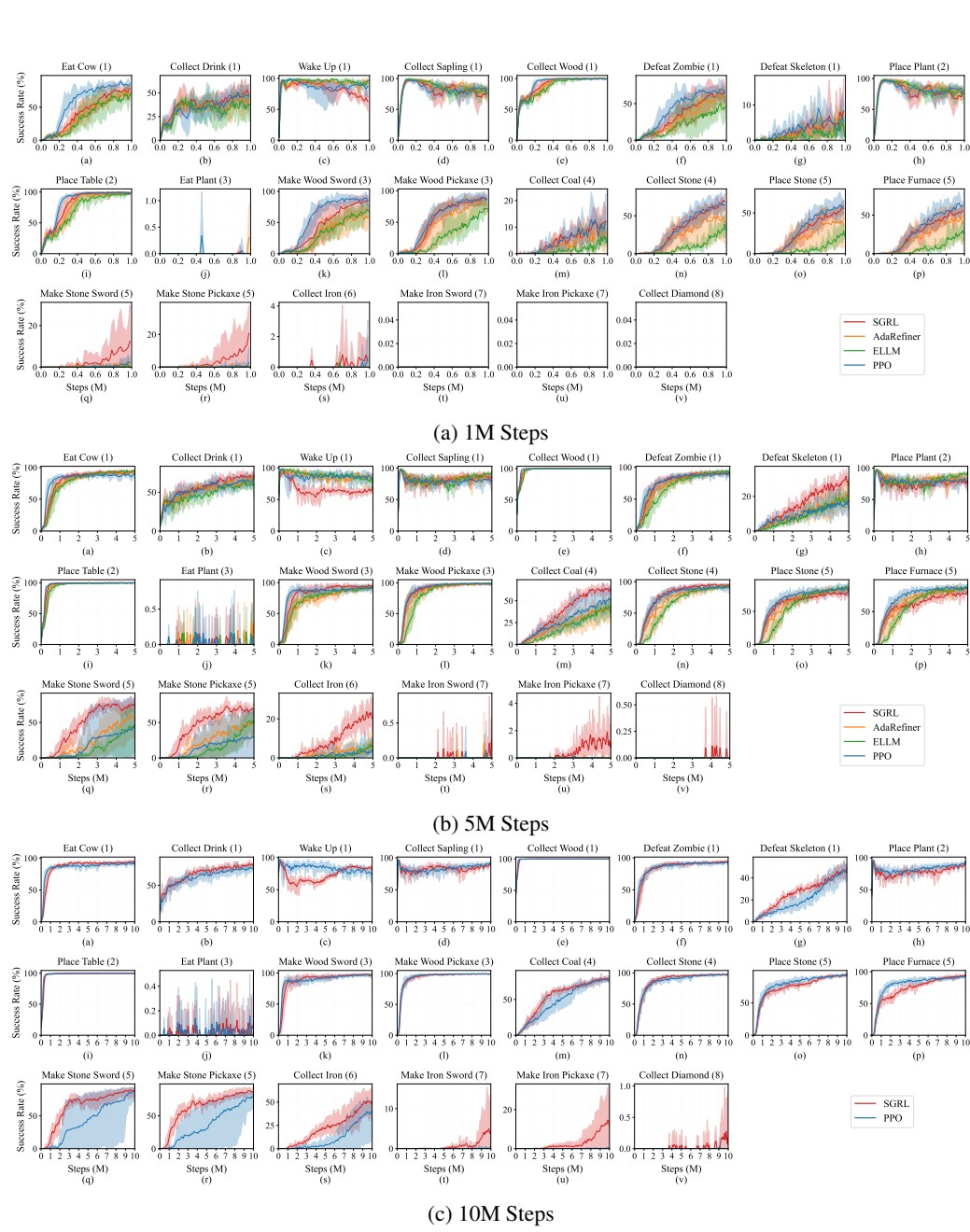

Figure 11: Main Results. Success rate curves for all achievements on Craftax-Classic within different training steps. We rank the achievements based on their depth and the importance of unlocking them for subsequent tasks. Achievements ranked later have greater depth and exert a stronger influence on subsequent achievements. A more intuitive version is shown in Figure 13.

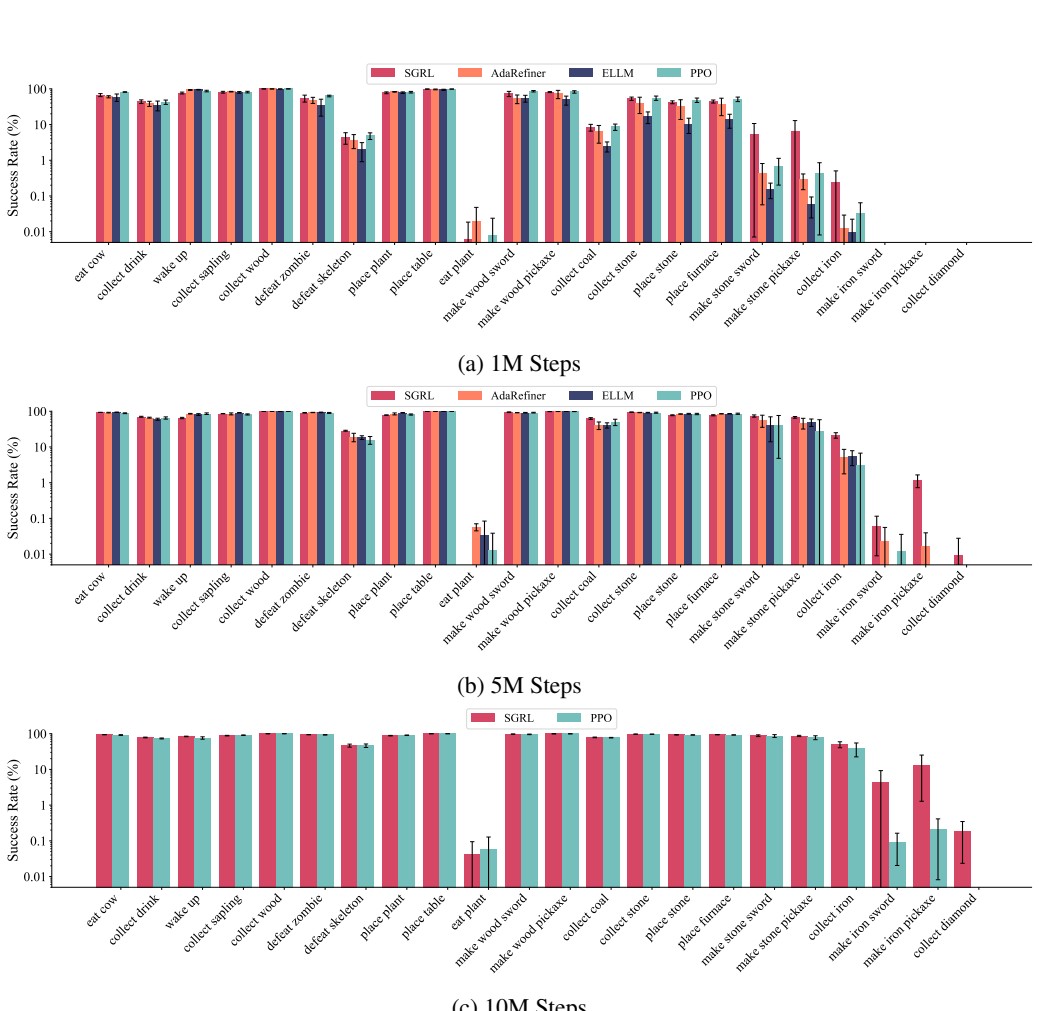

(a) 1M Steps

(b) 5M Steps

(c) 10M Steps

Figure 12: Main Results. Success rates across all achievements on Craftax-Classic at different training steps. **Note:** Since ELLM and AdaRefiner require frequent online calls to the LLM (`DeepSeek-V3`) during training, they incur substantial computational costs and training time. Therefore, we only reproduce the results within 5M steps.

| Method | Score(%) | Reward | Achievement Depth | SPS ($\times 10^2$) |
|---|---|---|---|---|
| Human | $50.5 \pm 6.8$ | $14.3 \pm 2.3$ | 8 | - |
| SGRL | $\mathbf{30.5} \pm 1.2$ | $12.7 \pm 0.4$ | **8** | 1.0 |
| AdaRefiner | $28.2 \pm 1.8$ | $\mathbf{12.9} \pm 1.2$ | 7 | - |
| ELLM | - | $6.0 \pm 0.4$ | - | - |
| PPO | $18.5 \pm 6.1$ | $10.1 \pm 1.3$ | 6 | 1.2 |

Table 4: Main Results. Performance of SGRL and baseline methods on Crafter at 5M steps.

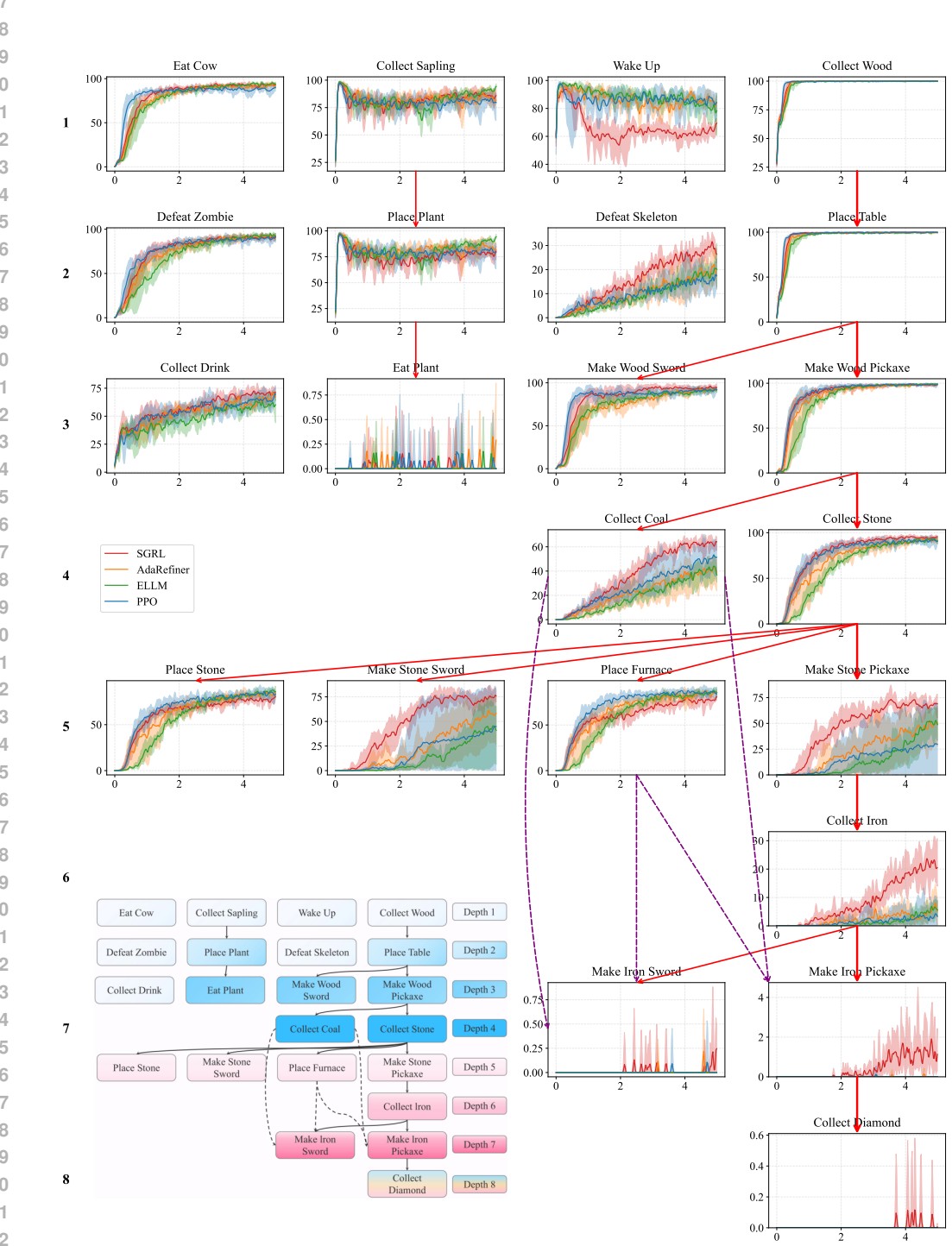

Figure 13: Main Results. Success rate curves for all achievements on Craftax-Classic within 5M steps. Solid and dashed arrows indicate direct and cross-depth dependencies, respectively. The bottom-left panel visualizes the full achievement dependency graph, with achievement depth encoded by color (depth 1-8 from top to bottom).

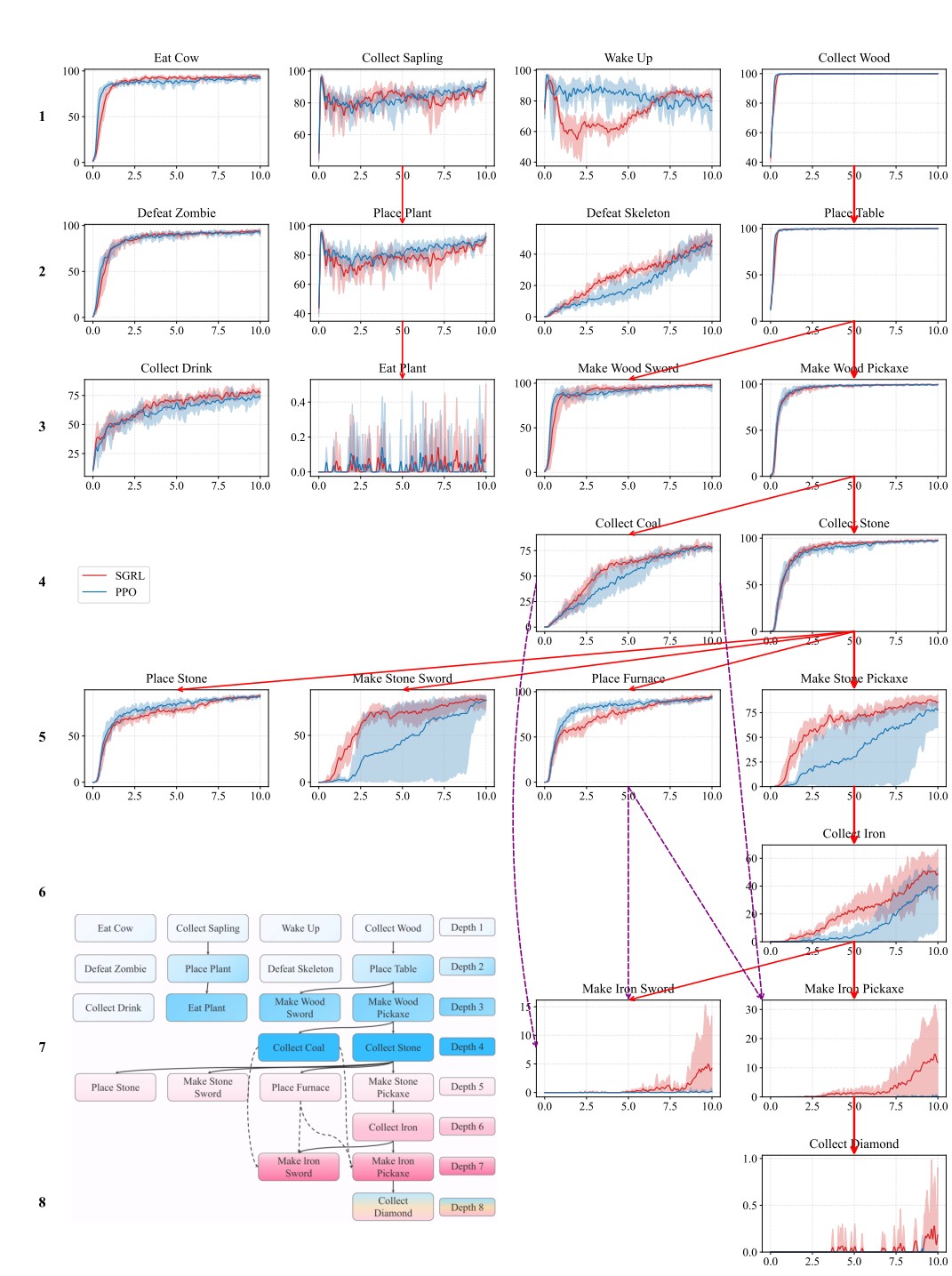

Figure 14: Main Results. Success rate curves for all achievements on Craftax-Classic within 10M steps. Solid and dashed arrows indicate direct and cross-depth dependencies, respectively. The bottom-left panel visualizes the full achievement dependency graph, with achievement depth encoded by color (depth 1-8 from top to bottom).

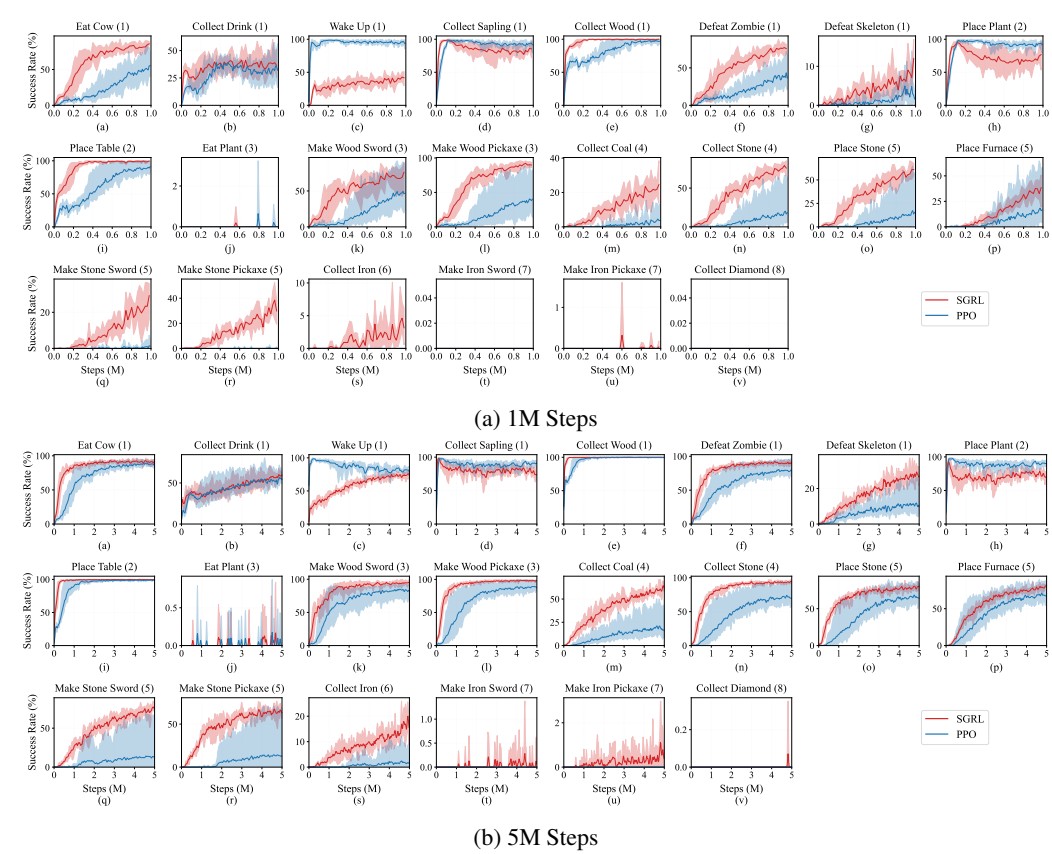

Figure 15: Main Results. Success rate curves for all achievements on Crafter within different training steps. We rank the achievements based on their depth and the importance of unlocking them for subsequent tasks. Achievements ranked later have greater depth and exert a stronger influence on subsequent achievements. A more intuitive version is shown in Figure 17.

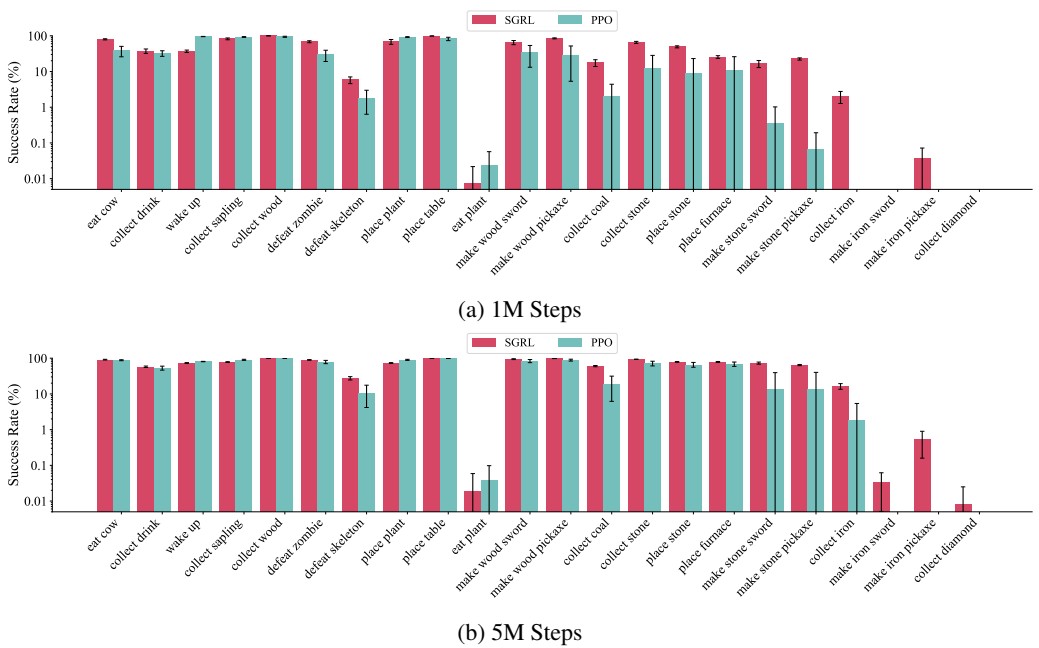

Figure 16: Main Results. Success rates across all achievements on Crafter at different training steps.

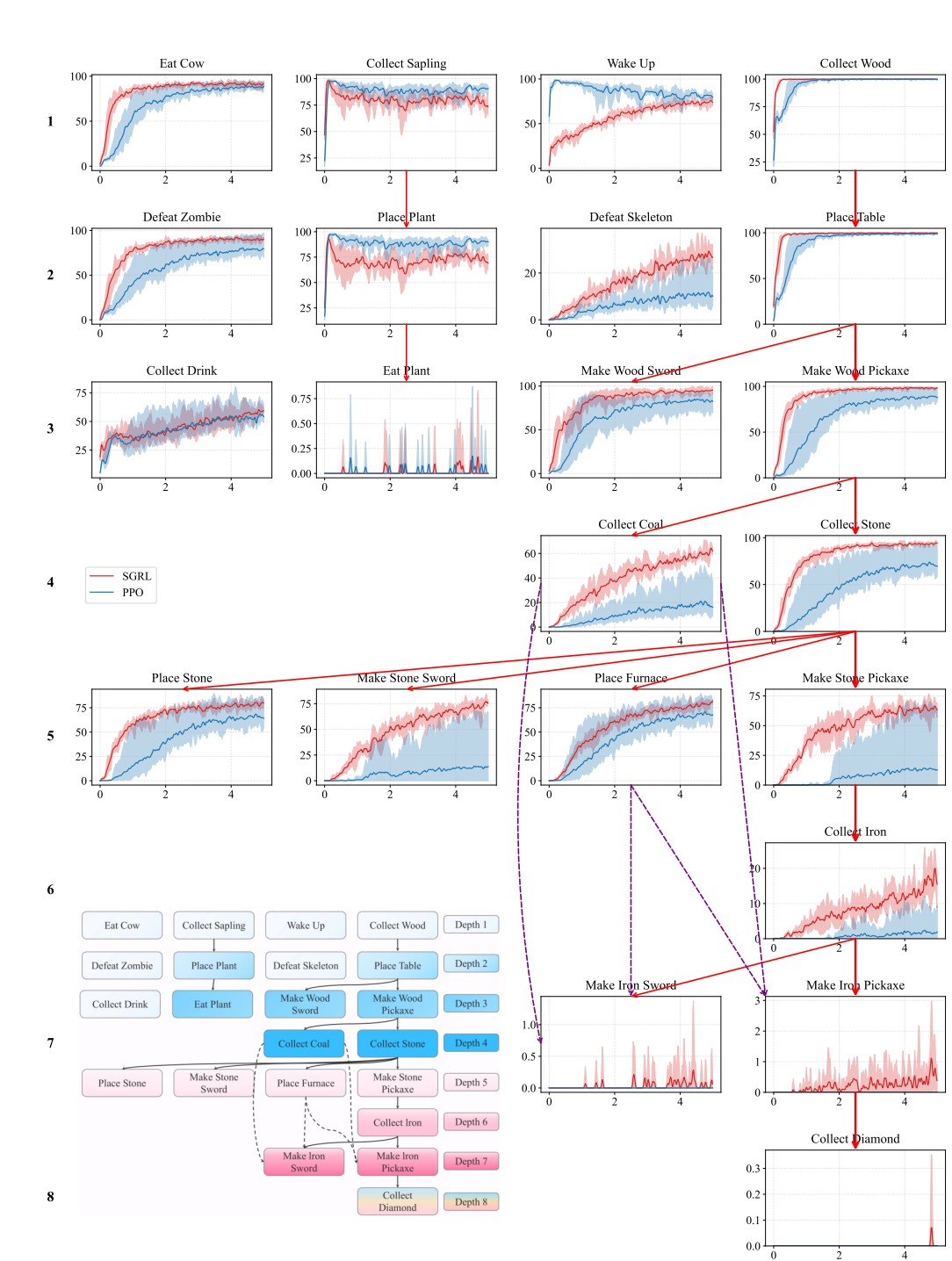

Figure 17: Main Results. Success rate curves for all achievements on Crafter within 5M steps. Solid and dashed arrows indicate direct and cross-depth dependencies, respectively. The bottom-left panel visualizes the full achievement dependency graph, with achievement depth encoded by color (depth 1-8 from top to bottom).

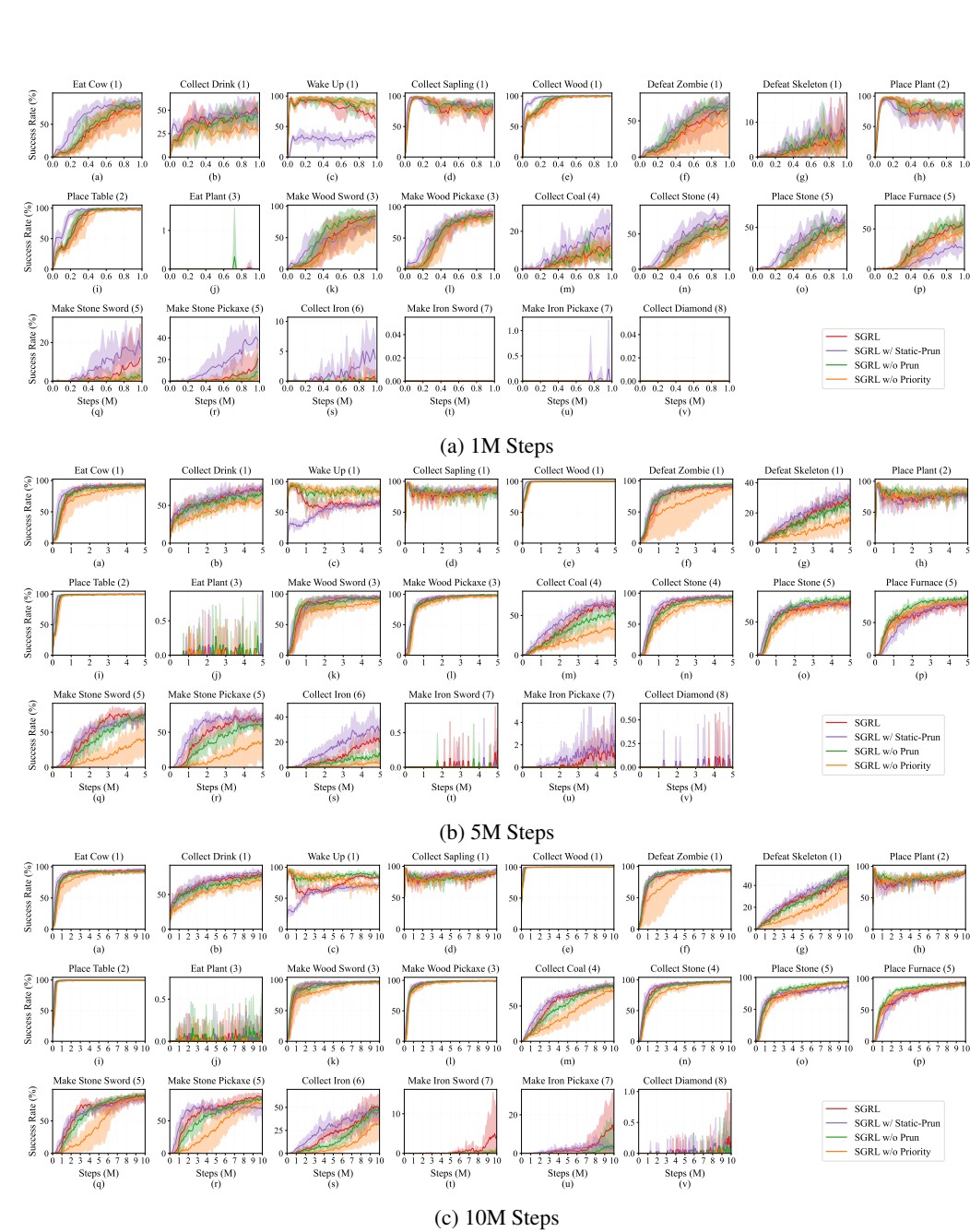

Figure 18: Ablation Studies. Success rate curves for all achievements on Craftax-Classic within different training steps. We rank the achievements based on their depth and the importance of unlocking them for subsequent tasks. Achievements ranked later have greater depth and exert a stronger influence on subsequent achievements. A more intuitive version is shown in Figure 20.

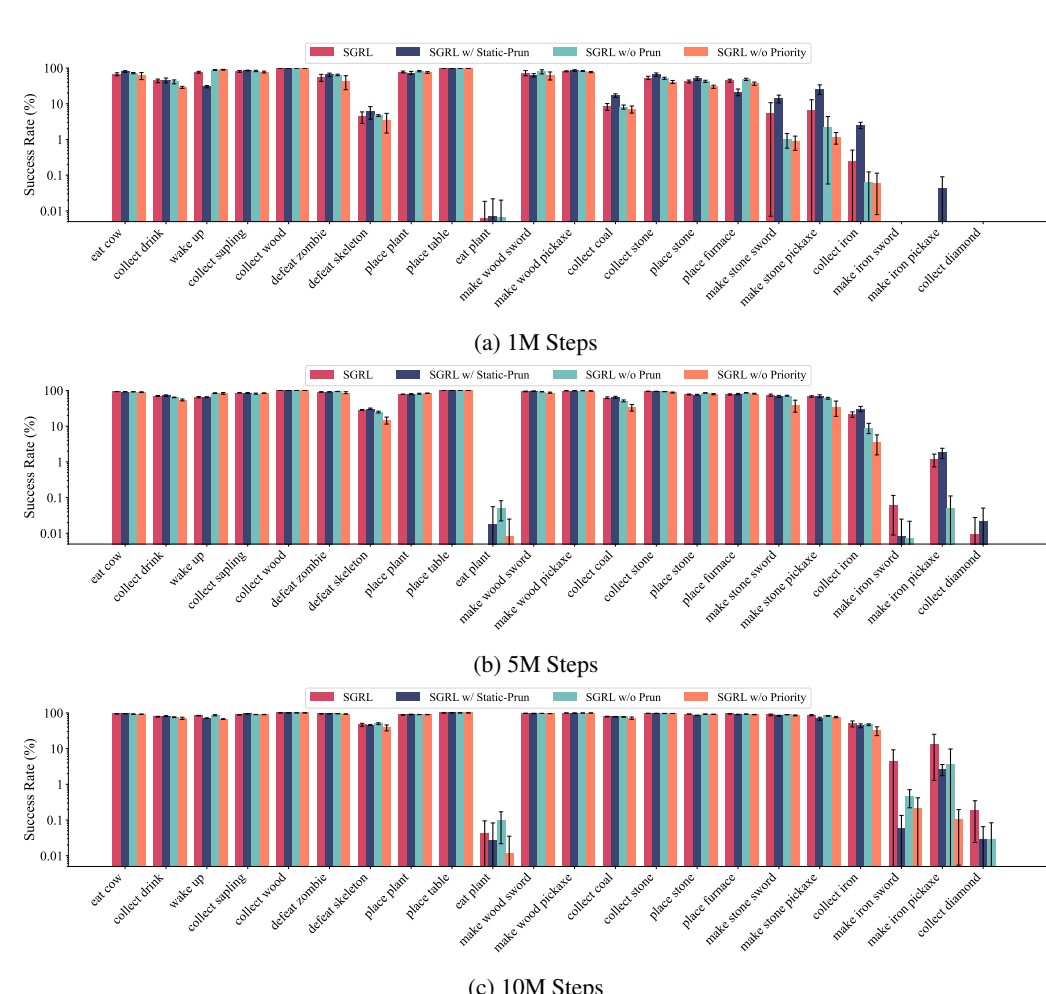

(a) 1M Steps

(b) 5M Steps

(c) 10M Steps

Figure 19: Ablation Studies. Success rates across all achievements on Craftax-Classic at different training steps.

| Method | Score(%) | Reward | Achievement Depth |
|---|---|---|---|
| SGRL | $33.8 \pm 1.5$ | $\mathbf{13.0} \pm 0.3$ | 8 |
| SGRL w/ Static-Prun | $\mathbf{34.2} \pm 1.7$ | $12.9 \pm 0.4$ | 8 |
| SGRL w/o Prun | $30.9 \pm 1.3$ | $12.7 \pm 0.2$ | 7 |
| SGRL w/o Priority | $30.4 \pm 0.9$ | $12.3 \pm 0.5$ | 7 |

Table 5: Ablation Studies. Performance of SGRL and ablation methods on Craftax-Classic at 5M steps.

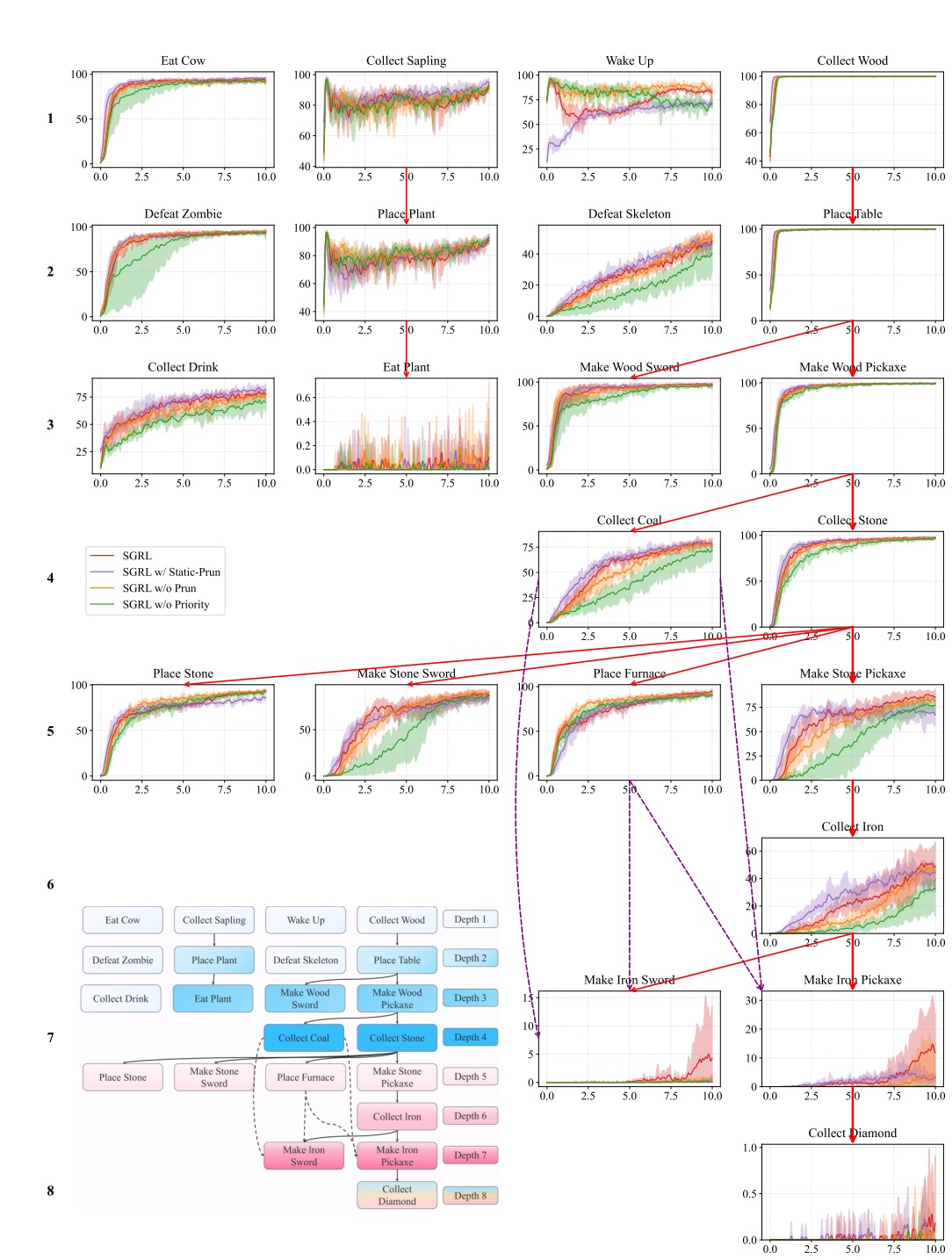

Figure 20: Ablation Studies. Success rate curves for all achievements on Craftax-Classic within 5M steps. Solid and dashed arrows indicate direct and cross-depth dependencies, respectively. The bottom-left panel visualizes the full achievement dependency graph, with achievement depth encoded by color (depth 1-8 from top to bottom).

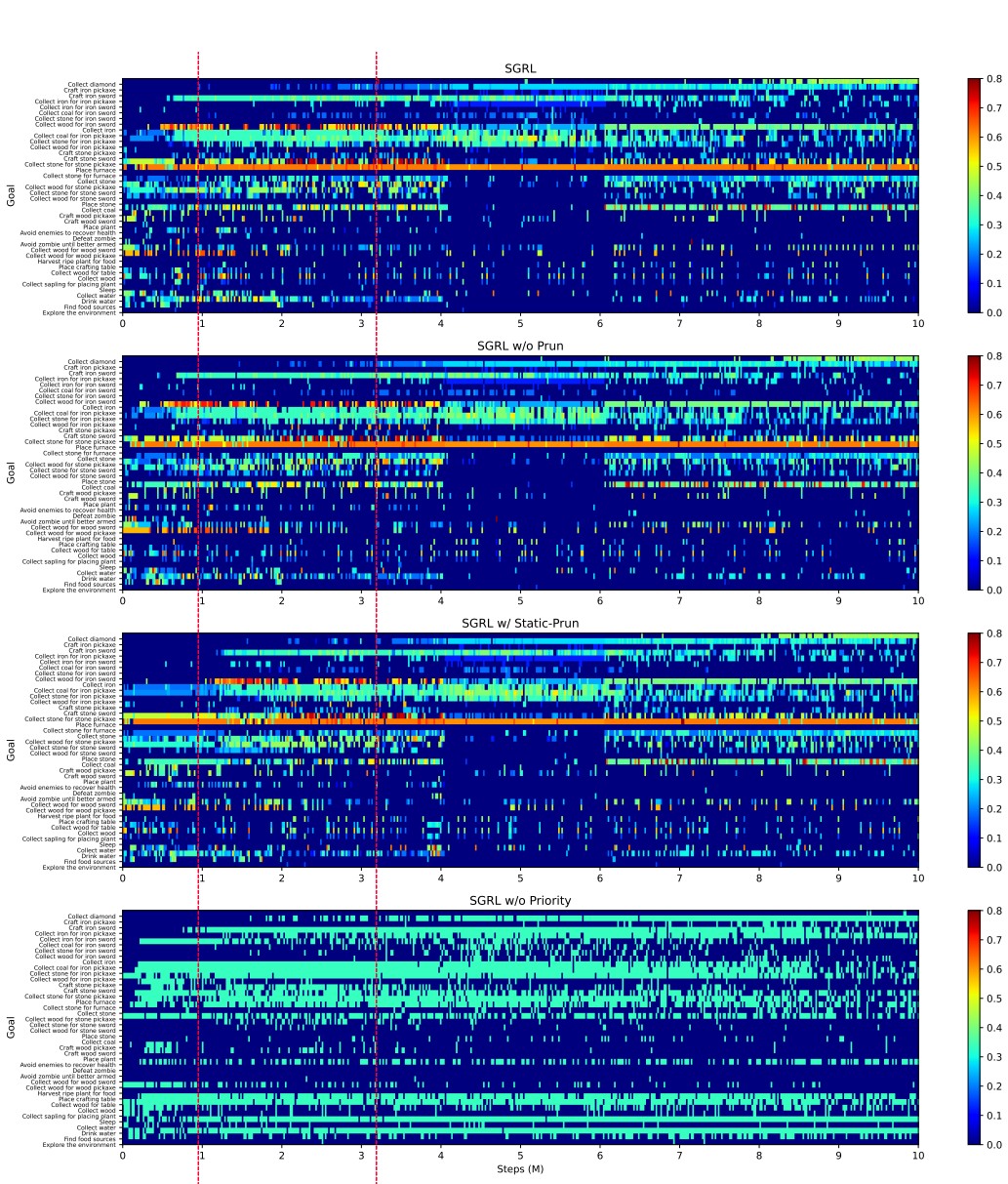

Figure 21: Ablation Studies. Heatmap of the goals with priority weights generated by the structured goal planner on Craftax-Classic within 10M steps. The vertical axis on the left shows goals ranked from low to high, while the right axis (ranging from 0 to 0.8) indicates the corresponding weights.

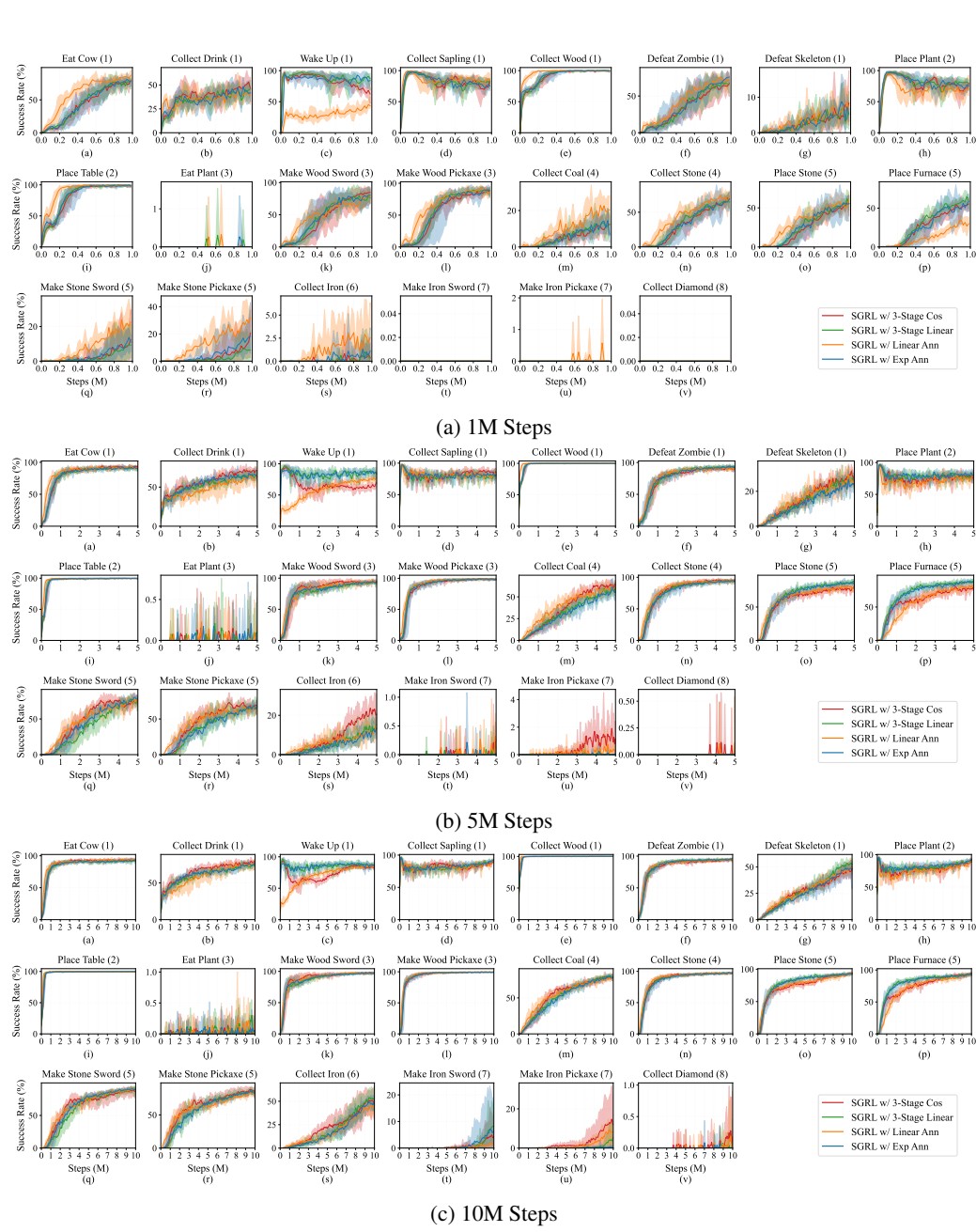

Figure 22: Mask Comparison. Success rate curves for all achievements on Craftax-Classic within different training steps. We rank the achievements based on their depth and the importance of unlocking them for subsequent tasks. Achievements ranked later have greater depth and exert a stronger influence on subsequent achievements. A more intuitive version is shown in Figure 24.

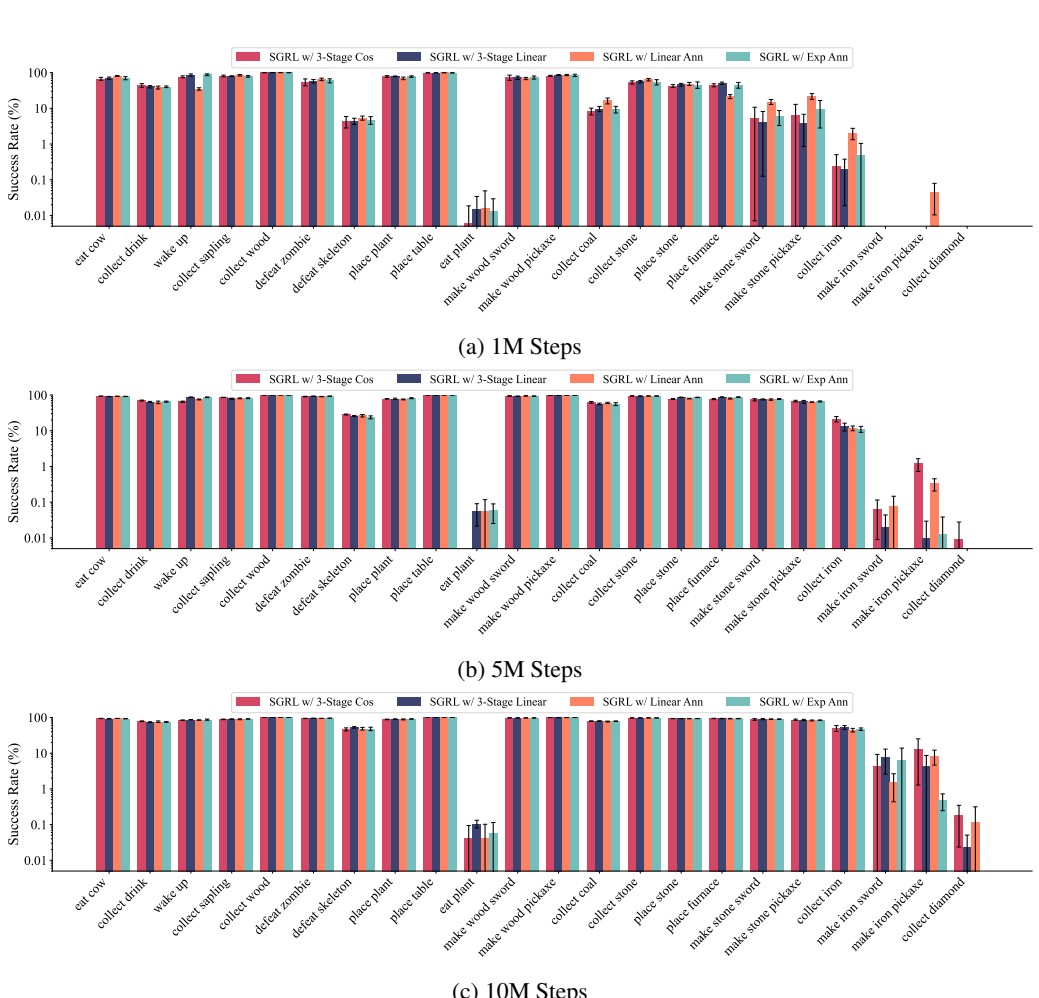

Figure 23: Mask Comparison. Success rates across all achievements on Craftax-Classic at different training steps.

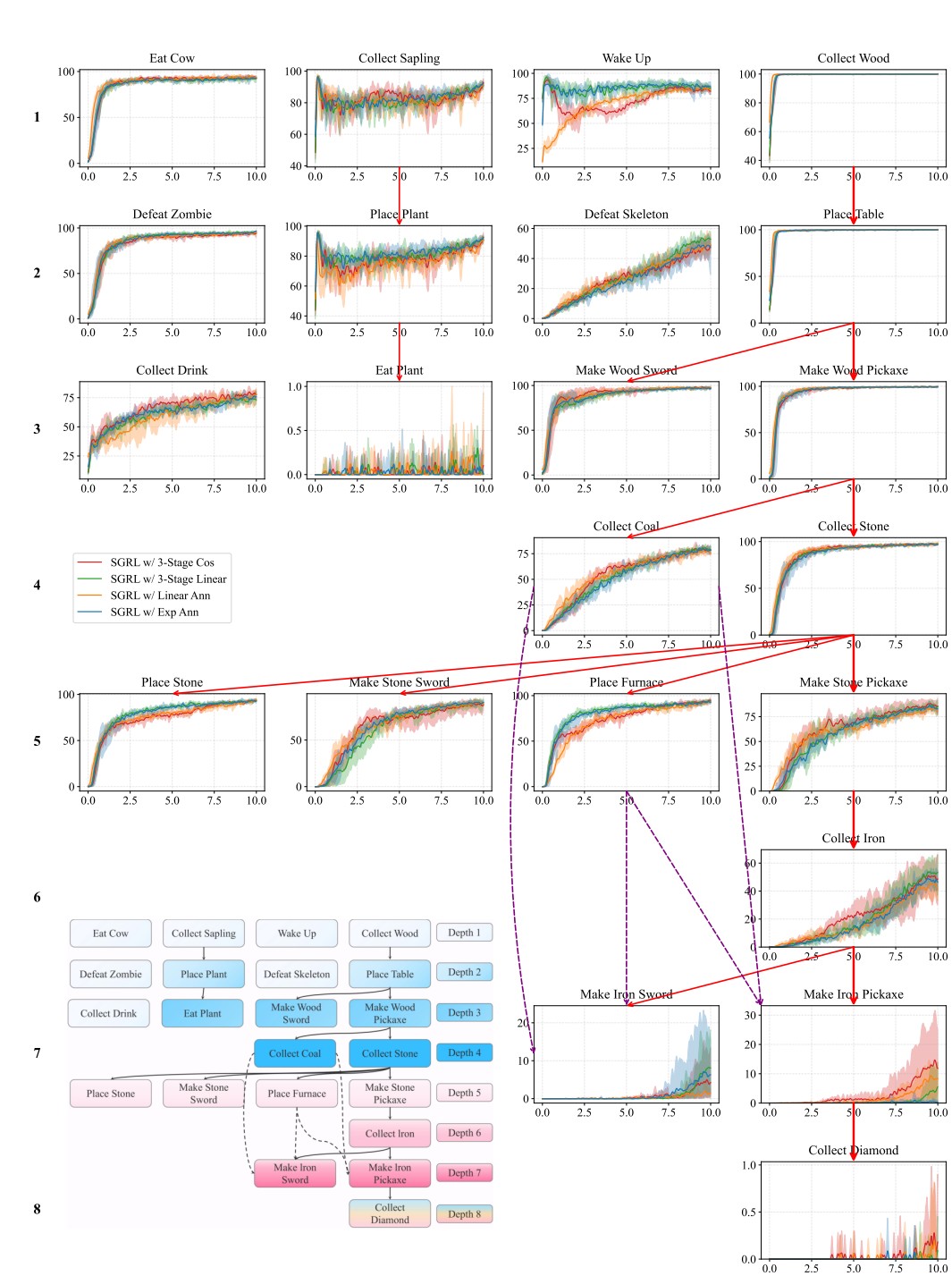

Figure 24: Mask Comparison. Success rate curves for all achievements on Craftax-Classic within 5M steps. Solid and dashed arrows indicate direct and cross-depth dependencies, respectively. The bottom-left panel visualizes the full achievement dependency graph, with achievement depth encoded by color (depth 1-8 from top to bottom).

