# OpenReview forum: "Goal-Guided Efficient Exploration via Large Language Model in Reinforcement Learning"
_ICLR.cc/2026/Conference — ICLR 2026 Conference Withdrawn Submission_

### Official Review · Reviewer_T3SB · 2025-10-28

**Soundness:** 2
**Presentation:** 2
**Contribution:** 2
**Rating:** 2
**Confidence:** 2

**Summary:**

This paper proposes SGRL (Structured Goal-guided Reinforcement Learning), which uses an LLM twice to generate a reusable, structured goal–generation function with priority weights. They also use LLM to produce a goal-conditioned action mask that constrains the PPO policy via logit masking with an annealed stochastic relaxation. The method is evaluated on Crafter and Craftax-Classic, with ablations on the pruner and the priority mechanism. The main claim is improved long-horizon exploration with fewer LLM calls than prior LLM-guided RL baselines.

**Strengths:**

1. The paper decouples LLM usage from per-step decision-making by compiling it into code (goal function) and banked masks, in spirit similar to code-as-interface approaches. The framework in Figure 2 is straightforward to map to implementation.

2. The paper includes several ablation variants and different annealing schedules, with comprehensive success-rate plots across all 22 achievements.

3. Prompt templates for both the goal planner and the pruner are provided, which is helpful for reproducibility.

**Weaknesses:**

1. The manuscript is very difficult to read: inconsistent notation (planning step h vs. timestep t in Sec. 2.2 vs. Sec. 3); dense, crowded figures with tiny fonts; and many copy-editing artifacts (hyphenation, spacing, and broken words throughout). For instance, the overall diagram in Figure 2 (page 3) mixes code blocks, prompts, and boxes with little visual hierarchy; important interfaces are not labeled precisely (e.g., how `g_emb` is computed and fed to the policy).

2. The mathematical exposition mixes on-policy and buffer language (Eq. 6: "replay buffer or on-policy rollout distribution D") in a PPO setting, which is confusing; masking is injected via a large negative constant C in Eq. 5 without a principled stability discussion.

3. The text states "SGRL requires only minimal LLM invocation, resulting in faster training speed," yet Table 1 shows PPS/SPS: PPO 135.3 vs. SGRL 18.5, meaning SGRL is much slower than plain PPO (though faster than ELLM/AdaRefiner). The paper should qualify “faster” as relative to other LLM-based baselines, not absolute.

4. The comparison set omits strong non-LLM exploration baselines on Crafter/Craftax (e.g., world-model or intrinsic-motivation methods). Since SGRL's pitch is “efficient exploration,” comparisons solely against PPO and two LLM-goal methods are insufficient to establish significance in the RL literature. The authors themselves note ELLM/AdaRefiner were only partially reproduced due to cost, and original Crafter numbers are copied from papers, further weakening fairness.

5. The approach leans heavily on an environment-specific text interface. The appendix includes a concrete function `render_craftax_text_describ_2` that converts internal map arrays and IDs into rich textual observations (pages 15–16), e.g., enumerating block/mob names within the agent’s view. This "oracle-like" textual channel could advantage goal generation in ways not available to pixels-only agents; fairness vs. PPO/other baselines is unclear, because the RL policy appears to use pixels while the planner sees high-level text (Figure 2(c)). Please clarify whether baselines had access to identical textual summaries.

**Questions:**

1. Text channel fairness: Did PPO and the LLM baselines receive the same textual observation stream that the goal planner uses (Appendix B.3, pages 15–16)? If not, please justify and report numbers with matched inputs.

2. How many LLM calls and tokens are used per 1M steps for (i) goal-code synthesis, (ii) priority updates, and (iii) mask generation?

3. Can the same goal code and mask bank transfer to a different open-ended environment (e.g., different crafting graphs) without re-prompting? What breaks?

---

### Official Review · Reviewer_SFBk · 2025-10-30

**Soundness:** 1
**Presentation:** 1
**Contribution:** 1
**Rating:** 0
**Confidence:** 4

**Summary:**

This paper proposes to use LLMs to generate a Python function that acts as goal planner and goal pruner in order to guide the exploration of the agent thanks to the LLM knowledge. The approach is evalauted on Crafter-Classic and Crafter.

**Strengths:**

The paper aims to improve exploration in reinforcement learning by leveraging LLM knowledge, which is a well known issue in the literature.

**Weaknesses:**

The paper does not contain sufficient details to explain the methodology.

The paper's most significant idea seems the shift from an online guidance paradigm, which is very expensive, to one where the LLM's knowledge is compiled "offline" into an executable and reusable function. However, the authors never explicitly frame their contribution this way. A more direct framing would improve the paper's clarity and better position the work within the broader context of program synthesis and neuro-symbolic methods.

The method's success appears to be heavily dependent on extremely detailed, domain-specific prompt engineering. The appendix reveals prompts that contain not just high-level task descriptions but also exhaustive lists of items, achievements, and even the source code for the environment's text rendering function. This raises critical questions about the method's generalizability and the true cost of its application (no principles or methodology for constructing these prompts are provided or no generated function are reported in the paper so it is impossible to understand the value of the offline generation).

**Questions:**

Could you please quantify the human effort and computational resources required for the initial "compilation" phase? For instance, how many iterations of prompt refinement and code revision were necessary to generate a functional and effective goal planner?

How sensitive is the final performance to the frequency of the goal priority weight updates? Does less frequent updating lead to a graceful degradation in performance?

Can you provide more insight into the types of errors the LLM made when generating the action masks? How often were these masks overly restrictive or overly permissive, and how crucial was the annealing schedule in practice? How sensitive is the agent performance to those hyperparameters?

Minor
- Figure 1 is too small

---

### Official Review · Reviewer_sq7B · 2025-11-01

**Soundness:** 3
**Presentation:** 2
**Contribution:** 2
**Rating:** 4
**Confidence:** 4

**Summary:**

The paper proposes a method to efficiently use large language models (LLMs) to guide goal-conditioned reinforcement learning (RL) agents in open-world environments. The key idea is to distill LLM knowledge by prompting it to generate a goal-weighting function $\phi$ as code. This function assigns weights to candidate goals based on the current state. Goals are sampled accordingly, and $\phi$ is periodically updated.

The resulting framework, Structured Goal-guided Reinforcement Learning (SGRL), is evaluated on Crafter and Craftax-Classic, showing improved exploration and performance over baselines that either exclude LLMs or rely on more frequent, costly LLM calls.

**Strengths:**

- Demonstrates clear and consistent performance gains over baselines such as PPO, ELLM, and AdaRefiner on both Crafter and Craftax-Classic benchmarks.
- Provides ablation studies showing the contribution of each component (goal prioritization and action pruning) to the final performance.
- Proposes a simple yet effective way to distill LLM knowledge into executable goal-generation code, substantially reducing the number of LLM calls during training while maintaining strong exploration efficiency.

**Weaknesses:**

- **Clarity and motivation of the main contribution**
The proposed approach builds on previous LLM–RL frameworks such as AdaRefiner and ELLM, focusing on reducing LLM call frequency through a goal-weighting function $\phi(s_t)$. While this is a practical direction, the conceptual motivation behind the “structured goal planner” could be made clearer. In particular, it remains uncertain why introducing goal prioritization (selecting multiple goals instead of one) should improve exploration beyond providing efficiency gains.
Including an ablation where $\phi$ outputs a single goal (without weighting) would help illustrate the specific benefit of the proposed mechanism.

- **Ambiguity in the definition and update of $\phi(s_t)$**
The paper inconsistently describes $\phi(s_t)$. Section 3.1 states it is “constructed at each timestep,” yet Appendix B.3.1 indicates that it is updated only every 2 million steps. This is a major clarity issue since $\phi$ is the core contribution.
It should be made explicit that:
  - $\phi$ is not recomputed at every step but only during periodic LLM update phases.
  - $\phi$ corresponds to executable code generated by the LLM and forms the main novelty of the approach.
The notion of “structured goal-generation function” is also vague and needs a clearer definition.

- **Unclear goal sampling and encoding process**
Several key implementation aspects are missing:
  - The paper never specifies how the priority weights are used to sample goals.
  - It remains unclear how the full goal–weight set $\{(g_t^i, w_t^i)\}$ is encoded into a single goal embedding vector — does this compress all goals into one representation, and how?
  - The differences between JAX and PyTorch implementations are mentioned but not explained, even though they can affect sampling behavior and reproducibility.

- **Restrictive evaluation setup**
The method is evaluated only on Crafter and Craftax-Classic, which are almost identical environments. These benchmarks are insufficient to support claims about generalization or exploration efficiency. More diverse and challenging environments (e.g., those used in ELLM or MineDojo) would be necessary to validate the approach.

- **Limited use of LLM capabilities**
Although the paper claims to “generate goals” with an LLM, the generated goals appear to be restricted to Crafter’s predefined achievements. In practice, the LLM mainly re-weights existing goals rather than producing novel or abstract ones, which limits both originality and th

- **Weakness of the ablation study**
The ablations are conducted on Craftax-Classic, where several baselines (ELLM, AdaRefiner) are not included. This makes it difficult to assess the relative contribution of each component compared to prior methods. Running ablations in the same setup as the main comparisons (e.g., on Crafter, where baselines are available) would provide a fairer and more interpretable evaluation of the proposed components.

**Questions:**

**Relation to curriculum learning.**
Your method leverages an LLM’s background knowledge to prioritize goals dynamically, guiding exploration toward “forward-looking” objectives. However, there is a rich literature on curriculum learning and automatic goal prioritization in reinforcement learning — for example, strategies that sample goals based on learning progress (i.e., the temporal derivative of success rate). Could you clarify how your approach relates to or differs from curriculum learning methods? (see https://arxiv.org/pdf/2003.04664 for a survey).
It might strengthen the paper to explicitly situate your approach within this line of work in the related-work section.

**On-policy consistency**
Since the goal-generation function $\phi$ and the corresponding goals may change during training, the resulting data distribution becomes non-stationary. This could effectively make training off-policy, whereas PPO is an on-policy algorithm. How do you handle or justify this apparent inconsistency?

---

### Official Review · Reviewer_uBcY · 2025-11-03

**Soundness:** 3
**Presentation:** 3
**Contribution:** 3
**Rating:** 4
**Confidence:** 4

**Summary:**

This paper introduces Structured Goal-guided Reinforcement Learning (SGRL), a novel approach that leverages Large Language Models (LLMs) to improve exploration efficiency in open-world reinforcement learning environments. Unlike existing methods that require frequent LLM invocations during training, SGRL generates a reusable, structured goal-generation function once and uses it to provide forward-looking goals with priority weights. The method consists of two main components: (1) a structured goal planner that creates parameterized goal-generation functions with dynamic priority weighting, and (2) a goal-conditioned action pruner that filters actions misaligned with current goals using an adaptive masking mechanism. The authors evaluate SGRL on Crafter and Craftax-Classic benchmarks, demonstrating superior performance compared to existing LLM-enhanced RL methods, particularly on long-horizon tasks requiring sequential planning.

**Strengths:**

1. The idea of using LLMs to generate structured, reusable goal-generation code rather than directly generating goals is innovative.
2. SGRL demonstrates clear improvements over baselines, particularly on challenging long-horizon achievements like "Collect Diamond" in Craftax-Classic. The method successfully unlocks deeper achievements that other methods fail to reach within the same training budget.

**Weaknesses:**

1. The evaluation is restricted to Crafter and Craftax-Classic, which are essentially variants of the same environment. The generalizability to other open-world environments (e.g., Minecraft, robotics tasks) remains unclear, limiting the broader impact of the work.
2. The evaluation is restricted to Crafter and Craftax-Classic, which are essentially variants of the same environment. The generalizability to other open-world environments (e.g., Minecraft, robotics tasks) remains unclear, limiting the broader impact of the work.
3. Due to computational constraints, some baseline methods (ELLM, AdaRefiner) could only be evaluated up to 5M steps, making it difficult to assess relative performance fairly across the full training horizon. This limitation weakens the comparative analysis.
4. The method involves multiple interacting components (goal generation, priority weighting, action masking, annealing schedules) that must be carefully tuned. The complexity may make it challenging to reproduce and adapt to new domains.

**Questions:**

1. How does SGRL perform in other types of open-world environments, such as navigation tasks, continuous control, or environments with different action spaces and observation modalities?
2. How sensitive is the method to the choice of LLM model? What happens when using smaller, less capable models, or when the LLM generates incorrect or suboptimal goal-generation code?
3. How does the method scale to environments requiring more complex goal hierarchies or longer dependency chains? What is the upper limit on the complexity of goals that can be effectively handled?
4. How sensitive is the method's performance to the specific prompts used for goal generation and action masking? Are there guidelines for designing effective prompts for new domains?

---

### Note · Authors · 2025-11-14

I have read and agree with the venue's withdrawal policy on behalf of myself and my co-authors.